# THE LIPSCHITZ CONSTANT OF SELF-ATTENTION

## ABSTRACT

Lipschitz constants of neural networks have been explored in various contexts in deep learning, such as provable adversarial robustness, estimating Wasserstein distance, stabilising training of GANs, and formulating invertible neural networks. Such works have focused on bounding the Lipschitz constant of fully connected or convolutional networks, composed of linear maps and pointwise non-linearities. In this paper, we investigate the Lipschitz constant of self-attention, a non-linear neural network module widely used in sequence modelling. We prove that the standard dot-product self-attention is *not* Lipschitz, and propose an alternative L2 self-attention that *is* Lipschitz. We derive an upper bound on the Lipschitz constant of L2 self-attention and provide empirical evidence for its asymptotic tightness. To demonstrate the practical relevance of our theoretical work, we formulate invertible self-attention and use it in a Transformer-based architecture for a character-level language modelling task.

## 1 INTRODUCTION

Lipschitz continuity is a strong form of continuity for functions. Loosely speaking, a function is *Lipschitz continuous* if changing its input by a certain amount cannot change its output by more than $K$ times that amount. The constant $K$ is a hard constraint on how rapidly the function's output can vary, and the smallest such $K$ is known as the function's *Lipschitz constant*. For example, $f_1(x) = \sqrt{|x|}$ and $f_2(x) = \exp(x)$ for $x \in \mathbb{R}$ are not Lipschitz continuous, because their output can change arbitrarily fast as $x$ approaches 0 and $+\infty$ respectively. On the other hand, $g_1(x) = \tanh(x)$ and $g_2(x) = \alpha x$ are Lipschitz continuous, because their rate of change (derivative) is bounded.

In deep learning, we often use Lipschitz continuity as a constraint for neural networks, to control how much a network's output can change relative to its input. Such Lipschitz constraints are useful in several contexts. For example, Lipschitz constraints can endow models with provable robustness against adversarial pertubations (Cisse et al., 2017; Tsuzuku et al., 2018; Anil et al., 2019), and guaranteed generalisation bounds (Sokolić et al., 2017). Moreover, the dual form of the Wasserstein distance is defined as a supremum over Lipschitz functions with a given Lipschitz constant, hence Lipschitz-constrained networks are used for estimating Wasserstein distances (Peyré & Cuturi, 2019). Further, Lipschitz-constrained networks can stabilise training for GANs, an example being spectral normalisation (Miyato et al., 2018). Finally, Lipschitz-constrained networks are also used to construct invertible models and normalising flows. For example, Lipschitz-constrained networks can be used as a building block for invertible residual networks and hence flow-based generative models (Behrmann et al., 2019; Chen et al., 2019). Additionally, Neural ODEs (Chen et al., 2018; Grathwohl et al., 2019) are typically defined using vector fields parameterized via Lipschitz networks, so that the flow generated by the vector field is guaranteed to exist for all times.

Nonetheless, designing Lipschitz-continuous neural networks and computing (or even upper-bounding) their Lipschitz constant is a hard problem. Previous work mostly focused on fully-connected and convolutional networks, not only because they are common in deep learning, but also because they are relatively simple to analyze, as compositions of linear maps and pointwise non-linearities. Even in this case however, exact evaluation of the Lipschitz constant of fully-connected and convolutional networks is NP-hard (Virmaux & Scaman, 2018) and obtaining a tight upper bound remains a challenging task (Virmaux & Scaman, 2018; Fazlyab et al., 2019; Latorre et al., 2020).

Fully-connected and convolutional networks are not the only neural networks worthy of interest. Recently, *self-attention* (Vaswani et al., 2017) has become a popular alternative to recurrent neural

networks. Self-attention is a key component of the Transformer (Vaswani et al., 2017), that has found success as a building block in models of various data modalities, starting with natural-language processing (Vaswani et al., 2017; Devlin et al., 2019; Brown et al., 2020) and extending to computer vision (Zhang et al., 2019; Parmar et al., 2019), audio generation (Huang et al., 2019), and reinforcement learning (Parisotto et al., 2020). However, so far no previous work has analyzed the Lipschitz properties of self-attention, and thus it has been unclear whether self-attention is a viable option in applications that require Lipschitz constraints. In this work, we address this gap in the theory of self-attention by providing a thorough analysis of its Lipschitz properties. In particular, we make the following contributions:

- We prove that the widely used *dot-product self-attention* is *not* Lipschitz, and therefore not suitable to use in applications requiring Lipschitz constraints.
- We formulate *L2 self-attention* as an alternative, and show that it *is* Lipschitz.
- We derive a theoretical upper bound on the Lipschitz constant of L2 self-attention, and provide empirical evidence of the asymptotic tightness of the bound.
- As a practical demonstration of the theory, we use this bound to formulate invertible self-attention, and explore its use in a Transformer architecture for character-level language modelling.

## 2 LIPSCHITZ CONSTANT OF FULLY-CONNECTED/CONVOLUTIONAL LAYERS

We first define the notion of Lipschitz continuity, and proceed to define the Lipschitz constant.

**Definition 2.1.** Given two metric spaces $(\mathcal{X}, d_{\mathcal{X}})$ and $(\mathcal{Y}, d_{\mathcal{Y}})$, a function $f : \mathcal{X} \to \mathcal{Y}$ is called *Lipschitz continuous* (or $K$-*Lipschitz*) if there exists a constant $K \geq 0$ such that

$$d_{\mathcal{Y}}(f(x), f(x')) \leq K d_{\mathcal{X}}(x, x') \quad \text{for all } x, x' \in \mathcal{X}. \tag{1}$$

The smallest such $K$ is the *Lipschitz constant* of $f$, denoted $\mathrm{Lip}(f)$.

In this paper, we focus on the common case where $\mathcal{X} = \mathbb{R}^n$, $\mathcal{Y} = \mathbb{R}^m$, and $d_{\mathcal{X}}, d_{\mathcal{Y}}$ are induced by a $p$-norm $\|x\|_p := (\sum_i |x_i|^p)^{1/p}$. We will primarily consider the cases $p = 2$ and $p = \infty$, where $\|x\|_\infty := \max_i |x_i|$. To emphasise the dependence of the Lipschitz constant on the choice of $p$-norm, we will often denote it by $\mathrm{Lip}_p(f)$. In this case, it follows directly from Definition 2.1 that the Lipschitz constant is given by

$$\mathrm{Lip}_p(f) = \sup_{x \neq x' \in \mathbb{R}^n} \frac{\|f(x) - f(x')\|_p}{\|x - x'\|_p}. \tag{2}$$

Next, we outline some basic results that are useful for estimating Lipschitz constants, also covered in related works (Virmaux & Scaman, 2018; Behrmann et al., 2019). We describe how these results are used to provide bounds on the Lipschitz constant of fully-connected networks (FCN) and convolutional neural networks (CNN), using the fact that both are compositions of linear maps and pointwise non-linearities. To begin with, the following theorem suggests a way to bound $\mathrm{Lip}_p(f)$ for a differentiable Lipschitz function $f$:

**Theorem 2.1** (Federer, 1969). *Let $f : \mathbb{R}^n \to \mathbb{R}^m$ be differentiable and Lipschitz continuous under a choice of $p$-norm $\| \cdot \|_p$. Let $J_f(x)$ denote its total derivative (Jacobian) at $x$. Then $\mathrm{Lip}_p(f) = \sup_{x \in \mathbb{R}^n} \|J_f(x)\|_p$ where $\|J_f(x)\|_p$ is the induced operator norm on $J_f(x)$.*

Hence if $f$ is a linear map represented by a matrix $W$ then

$$\mathrm{Lip}_p(f) = \|W\|_p := \sup_{\|x\|_p = 1} \|Wx\|_p = \begin{cases} \sigma_{\max}(W), & \text{if } p = 2 \\ \max_i \sum_j |W_{ij}| & \text{if } p = \infty \end{cases} \tag{3}$$

where $\|W\|_p$ is the operator norm on matrices induced by the vector $p$-norm, and $\sigma_{\max}(W)$ is the largest singular value of $W$. Under this choice of norm, many common non-linearities (including `relu`, `sigmoid`, `tanh`, `elu`) are 1-Lipschitz. $\|W\|_2 = \sigma_{\max}(W)$ is usually estimated via *power iteration*; we provide details on how this is done in Appendix B.

Since we now know the Lipschitz constants of the components of both FCN and CNN, we can bound their Lipschitz constants by applying the following lemma:

**Lemma 2.1** (Federer, 1969). *Let $g, h$ be two composable Lipschitz functions. Then $g \circ h$ is also Lipschitz with $\mathrm{Lip}(g \circ h) \leq \mathrm{Lip}(g) \mathrm{Lip}(h)$.*

**Corollary 2.1.** For a fully-connected network (FCN) or a convolutional neural network (CNN) $f = W_K \circ \rho_{K-1} \circ W_{K-1} \circ \ldots \circ \rho_1 \circ W_1$, we have $\mathrm{Lip}_p(f) \leq \prod_k \|W_k\|_p$ under a choice of $p$-norm with 1-Lipschitz non-linearities $\rho_k$.

The above bound is not necessarily tight; there are various works that compute tighter bounds for FCN and CNN (e.g. Virmaux & Scaman, 2018; Fazlyab et al., 2019; Latorre et al., 2020).

## 3 LIPSCHITZ CONSTANT OF SELF-ATTENTION

### 3.1 DOT-PRODUCT SELF-ATTENTION IS *not* LIPSCHITZ

Moving on, we investigate whether self-attention is Lipschitz. We first consider the widely used *(scaled) dot-product multihead self-attention* as formulated by Vaswani et al. (2017). Let $x_1, \ldots, x_N$ be a sequence of $N$ elements, where $x_i \in \mathbb{R}^D$ for $i = 1, \ldots, N$. We represent this sequence as a matrix $X \in \mathbb{R}^{N \times D}$ such that the $i$th row of $X$ is the $i$th element of the sequence, i.e. $X_{i:} = x_i^\top$. Dot-product multihead self-attention (DP-MHA) is a map from $\mathbb{R}^{N \times D}$ to $\mathbb{R}^{N \times D}$ consisting of $H$ 'heads', where $H$ is chosen to divide $D$. Each head is a map from $\mathbb{R}^{N \times D}$ to $\mathbb{R}^{N \times D/H}$ defined by

$$DP(X) := \mathrm{softmax}\left(XW^Q(XW^K)^\top / \sqrt{D/H}\right) XW^V, \tag{4}$$

where $W^Q, W^K, W^V \in \mathbb{R}^{D \times D/H}$ are learnable parameters specific to each head. The input to the softmax is an $N \times N$ matrix of dot products (hence *dot-product* self-attention), and the softmax is applied to each row of this matrix. Finally, the outputs of all heads are concatenated into an $N \times D$ matrix and are right multiplied by $W^O \in \mathbb{R}^{D \times D}$, thus DP-MHA is defined by

$$MHA(X) := \left[DP^1(X), \ldots, DP^H(X)\right] W^O. \tag{5}$$

In what follows, we will prove that $MHA$ as defined above is *not* Lipschitz, assuming that the $MHA$ map is non-trivial, i.e. $W^Q, W^K, W^V, W^O \neq 0$. It is sufficient to show that a single head $DP$ is not Lipschitz, since $MHA$ is a linear combination of the outputs of each head. Let us write Equation (4) as $DP(X) = PXW^V$, where $P \in \mathbb{R}^{N \times N}$ is the output of the softmax (we suppress the dependence of $P$ on $X$ to reduce clutter below). $P$ is a stochastic matrix, i.e. its entries are non-negative and its rows sum to 1. Since the rows of $X$ are the $x_i$'s, a linear transformation of each $x_i$ by some matrix $A$ is equivalent to right multiplication of $X$ by $A^\top$. So right multiplication of $X$ by $W^V$ is a linear map and thus Lipschitz. Therefore, we are interested in the mapping $f(X) = PX$; this is *not* a linear mapping because $P$ itself is a non-linear function of $X$. In fact, we show that $f$ is *not* Lipschitz, thus proving the first main result of the paper:

**Theorem 3.1.** DP-MHA *is not Lipschitz for any vector $p$-norm $\|\cdot\|_p$ with $p \in [1, \infty]$.*

*Summary of Proof.* We use Theorem 2.1, noting that if the supremum of the norm of the Jacobian is infinite, then the mapping is not Lipschitz. In particular, we show that when $x_i = 0$ for some $i$, some elements of the Jacobian of $f$ grow proportionally to the sample variance of $x_{\neq i}$, which is unbounded.

*Proof.* We show the proof for the case $D = 1$ (i.e. $X \in \mathbb{R}^{N \times 1}$, a column vector) for readability. See Appendix C for the general case, which follows the same logic.

The mapping $f$ can be written as

$$f(X) = PX = \mathrm{softmax}\left(aXX^\top\right)X = \begin{bmatrix} f_1(X)^\top \\ \vdots \\ f_N(X)^\top \end{bmatrix} \in \mathbb{R}^{N \times 1}, \tag{6}$$

where $a = W^K W^Q \in \mathbb{R}$ (we assume $a \neq 0$ such that self-attention is non-trivial) and $f_i(X) = \sum_{j=1}^N P_{ij} x_j$ with $P_{i:} = \mathrm{softmax}(ax_i X)$. Hence $f$ can be interpreted as a map of each $x_i$ to a point in the convex hull of $x_1, ..., x_N$. Since $f$ is a map from $\mathbb{R}^{N \times 1}$ to $\mathbb{R}^{N \times 1}$, its Jacobian is

$$J_f = \begin{bmatrix} J_{11} & \ldots & J_{1N} \\ \vdots & \ddots & \vdots \\ J_{N1} & \ldots & J_{NN} \end{bmatrix} \in \mathbb{R}^{N \times N}, \tag{7}$$

where $J_{ij} = \frac{\partial f_i(X)}{\partial x_j} \in \mathbb{R}$. By taking partial derivatives we can show that $J_{ij} = aX^\top P^{(i)} [e_{ji} X + \delta_{ij} X] + P_{ij} I$ where $e_{ij} \in \mathbb{R}^{N \times N}$ is a binary matrix with zeros everywhere

except the $(i, j)$th entry, $\delta_{ij}$ is the Kronecker delta, and $P^{(i)} := \text{diag}(P_{i:}) - P_{i:}^\top P_{i:}$. So for $i = j$:

$$J_{ii} = aX^\top P^{(i)} e_{ii} X + aX^\top P^{(i)} X + P_{ii} \tag{8}$$

Let us investigate the scalar $X^\top P^{(i)} X$. We observe that it is in fact a variance of a discrete distribution. Specifically:

$$X^\top P^{(i)} X = \sum_k P_{ik} x_k^2 - \left(\sum_k P_{ik} x_k\right)^2 = \text{Var}(\mathbb{X}), \tag{9}$$

where $\mathbb{X}$ is a discrete distribution with support at the inputs $\{x_1, \ldots, x_N\}$ and probability mass function given by their softmax probabilities $\mathbb{P}(\mathbb{X} = x_j) = P_{ij}$. A consequence of this interpretation is that $P^{(i)}$ is *positive semi-definite* (PSD) since $X^\top P^{(i)} X = \text{Var}(\mathbb{X}) \geq 0$, with equality if and only if the $x_j$ are all equal.

We use this observation to show that $J_{ii}$ is unbounded, and so $\|J_f\|_p$ is unbounded, hence `DP-MHA` is *not* Lipschitz. Consider the case $x_i = 0$. Then $P_{i:}^\top = \text{softmax}(XAx_i) = \frac{1}{N}\mathbb{1}$, i.e. we have uniform attention regardless of $x_{\neq i}$. The first term of $J_{ii}$ in Equation (8) disappears since $e_{ii} X = [0, \ldots, x_i, \ldots, 0] = \mathbf{0}$, and the last term becomes $\frac{1}{N} I$. Now consider the second term $aX^\top P^{(i)} X = a\text{Var}(\mathbb{X}_l)$. Note $\mathbb{X}$ is uniformly distributed, since $\mathbb{P}(\mathbb{X} = x_j) = P_{ij} = 1/N$. Hence the second term is equal to $a$ times the sample variance of $x_1, \ldots, x_N$, which can be arbitrarily large. $\square$

*High-level intuition for proof.* At $x_i = 0$, $f_i(X) = \frac{1}{N} \sum_k x_k$, the mean of the inputs. The rate of change of $f_i$ is governed by how fast the softmax saturates when $x_i$ is perturbed, which is determined by how spread out the $x_{\neq i}$ are. The more spread out they are (the higher the sample variance), the greater the rate of saturation of the softmax, and the faster the rate of change of $f_i$. Since the sample variance of $x_{\neq i}$ can be arbitrarily large, the rate of change of $f_i$ can also be arbitrarily large, i.e. the entries of the Jacobian (and hence its $p$-norm) can become arbitrarily large. In Appendix D, we show that adding bias terms to $x_i^\top W^Q$ and $x_j^\top W^K$ does *not* resolve the issue.

The implications of this result are the following. (1) There can be undesirable behaviour (e.g. training instabilities) for the Transformer when some inputs are close to zero. (2) Dot-product self-attention (and hence the standard Transformer) is not a suitable choice when we require a Lipschitz neural network, such as for formulating invertible residual networks (Behrmann et al., 2019). Therefore, to use self-attention and Transformers in such applications, a Lipschitz formulation of self-attention is required, together with an explicit (ideally tight) upper bound to its Lipschitz constant, to quantify how much the output can change with respect to changes in the input.

One method to make dot-product self-attention Lipschitz is by ensuring its inputs are bounded. Indeed, if the input space is compact, e.g. $[0, 1]^{N \times D}$, any continuously differentiable function is Lipschitz, including dot-product self-attention. However, as we further discuss in Section 6, such an approach has its own challenges, since it makes the Lipschitz constant depend on the input range. Instead, in the next section we formulate a version of self-attention that is provably Lipschitz on all of $\mathbb{R}^{N \times D}$, allowing us to derive an upper bound that holds for any subset of $\mathbb{R}^{N \times D}$.

## 3.2 L2 SELF-ATTENTION: A LIPSCHITZ FORMULATION OF SELF-ATTENTION

The pathology in dot-product self-attention arises because the softmax probabilities $P_{i:}$ are constant with respect to $x_{\neq i}$ when $x_i = 0$. This behaviour can be undesirable as we want $P_{ij}$ to vary according to $x_j$, regardless of whether $x_i$ is zero or not. Hence we propose an alternative form of self-attention based on L2 distance:

$$P_{ij} \propto \exp(L_{ij}) := \exp\left(-\left\|x_i^\top W^Q - x_j^\top W^K\right\|_2^2 / \sqrt{D/H}\right), \tag{10}$$

with the normalisation constant ensuring that $\sum_j P_{ij} = 1$. We will refer to it as *L2 self-attention*. It is reminiscent of the standard squared-exponential kernel, but with softmax normalisation that ensures that each row of the kernel matrix sums to 1. Normalisation is usually necessary to deal with inputs of varying length $N$ (Wang et al., 2018), hence we keep the softmax for L2 self-attention. Similarly to dot-product self-attention, L2 self-attention can be computed efficiently with matrix operations; see Appendix E for details, with a comparison of wall-clock runtimes between different choices of attention.

We first state the mathematical formulation of L2 multihead self-attention (`L2-MHA`) before proving the main result — the upper bound of its Lipschitz constant with respect to $\|\cdot\|_p$ for $p = 2, \infty$. The

full `L2-MHA` map $F : \mathbb{R}^{N \times D} \to \mathbb{R}^{N \times D}$ is defined as

$$F(X) := \left[ f^1(X) W^{V,1}, \ldots, f^H(X) W^{V,H} \right] W^O \quad \text{where} \quad f^h(X) := P^h X A_h.$$

In the above, $W^{V,h} \in \mathbb{R}^{D \times D/H}$, $W^O \in \mathbb{R}^{D \times D}$, $P^h$ is defined as in Equation (10) with $W^{Q,h} = W^{K,h} \in \mathbb{R}^{D \times D/H}$, and $A_h := W^{Q,h} W^{Q,h^\top} / \sqrt{D/H} \in \mathbb{R}^{D \times D}$. There are two changes from the usual form of multihead self-attention:

(1) We require $W^{Q,h} = W^{K,h}$ for each head $f^h(X)$ to be Lipschitz. In Lemma F.1 of Appendix F we show that `L2-MHA` is *not* Lipschitz for arbitrary $W^{Q,h}$, $W^{K,h}$, and that tying $W^{Q,h} = W^{K,h}$ is sufficient for `L2-MHA` to be Lipschitz, with intuition for why tying is sufficient.

(2) In each head of the self-attention $f^h(X)$, right multiplication by $A_h$ has been included for the theorem below to hold (details are in the proof). In practice, there is little harm done by this extra linear transformation, since when the heads are combined together in $F$, each $f^h(X)$ is additionally transformed by $W^{V,h}$, a free parameter.

The second main result of the paper is the following:

**Theorem 3.2.** `L2-MHA` *is Lipschitz, with the following bound on* $\text{Lip}_\infty(F)$:

$$\text{Lip}_\infty(F) \le \left( 4\phi^{-1}(N-1) + \frac{1}{\sqrt{D/H}} \right) \max_h \|W^{Q,h}\|_\infty \|W^{Q,h^\top}\|_\infty \max_h \|W^{V,h^\top}\|_\infty \|W^{O^\top}\|_\infty$$

*and the following bound on* $\text{Lip}_2(F)$:

$$\text{Lip}_2(F) \le \frac{\sqrt{N}}{\sqrt{D/H}} \left( 4\phi^{-1}(N-1) + 1 \right) \left( \sqrt{\sum_h \|W^{Q,h}\|_2^2 \|W^{V,h}\|_2^2} \right) \|W^O\|_2$$

*where* $\phi(x) := x \exp(x+1)$ *is an invertible univariate function on* $x > 0$, *and* $N$ *is the input sequence length.*

*Specifically,* $\phi^{-1}(N-1) = W_0(\frac{N}{e})$ *where* $W_0$ *is the Lambert* $W$-*function, which grows sublogarithmically as* $O(\log N - \log \log N)$ *(Corless et al., 1996). Hence the above bounds can be simplified to* $O(\log N)$ *for* $p = \infty$ *and* $O(\sqrt{N} \log N)$ *for* $p = 2$.

*Proof.* See Appendix F, which uses the key observation that $X^\top P^{(i)} X$ is a covariance matrix (c.f. Equation (9)) to bound $\|J_F\|_p$, the norm of the Jacobian of $F$. Appendix G shows how the argument can be modified to prove the analogous result for the case with masking in the self-attention. $\square$

These bounds are complemented by the concurrent work of Vuckovic et al. (2020), which provides a $O(\sqrt{D \log N})$ bound on $\text{Lip}_1(F)$ using measure-theoretic tools.

## 4 APPLICATION: INVERTIBLE SELF-ATTENTION

### 4.1 INVERTIBLE RESIDUAL NETWORK

Consider the residual function $g(x) := x + f(x)$. Behrmann et al. (2019) give the following sufficient condition for its invertibility: if $f$ is a *contraction* with respect to some metric, i.e. if $\text{Lip}(f) < 1$, and the metric space on which $f$ is defined is complete, then $g$ is invertible. (A Euclidean space with a metric induced by a $p$-norm $\| \cdot \|_p$ for $p \in [1, \infty]$ is always complete.) Specifically, the inverse $g^{-1}(y)$ is the unique fixed point of the recursion $x^{i+1} := y - f(x^i)$, since by the definition of the inverse we have $y = g^{-1}(y) + f(g^{-1}(y))$. Because $f$ is a contraction, *Banach's Fixed Point Theorem* guarantees that this fixed point exists and is unique for all $y$, and that the recursion converges for all initial values $x^0$ (often set to $y$ in practice) exponentially fast. Hence the inverse can be computed to arbitrary accuracy (up to numerical precision in practice) by the above fixed-point iteration.

Note that a composition of such invertible residual blocks is also invertible. Behrmann et al. (2019) use this observation to design invertible ResNets: they take $f$ to be a `CNN` normalised by an upper bound on $\text{Lip}(f)$ given by Corollary 2.1, making the resulting function *contractive*. For the 2-norm $\| \cdot \|_2$, a hyperparameter $c < 1$ is chosen and each linear map (convolution) $W$ in the `CNN` is multiplied by $c/\|W\|_2$ if $c < \|W\|_2$ where $\|W\|_2$ is estimated by power iteration (c.f. Appendix B). This multiplicative factor determines the scale of the Lipschitz constant of the normalised function.

## 4.2 INVERTIBLE SELF-ATTENTION

The standard use case of self-attention is with a skip connection inside the Transformer. A Transformer block is composed of residual blocks of multihead self-attention (MHA) and fully-connected (FCN) layers (Figure 1). Hence similarly to invertible ResNets, we can normalise `L2-MHA` by the upper bounds given in Theorem 3.2 to obtain `Contractive-L2-MHA` $f$, with which we can obtain invertible self-attention $g(x) = x + f(x)$.

In the next section, we investigate the properties of invertible self-attention and how it compares with the standard dot-product self-attention; we replace `DP-MHA` in the Transformer with `Contractive-L2-MHA`, hence replacing the residual self-attention module with invertible self-attention. We are not interested in the modified Transformer per se, but rather in comparing the properties of invertible self-attention to standard self-attention — we only use the Transformer as a testbed for this purpose, since self-attention is commonly used in a Transformer. Given the theoretical focus of the paper, we believe that a more challenging application of invertible self-attention, such as normalising flow-based modelling, would be more suitable as a separate paper focused on that particular application. In Appendix H, we show that `Dropout` in the residual branch is also contractive.

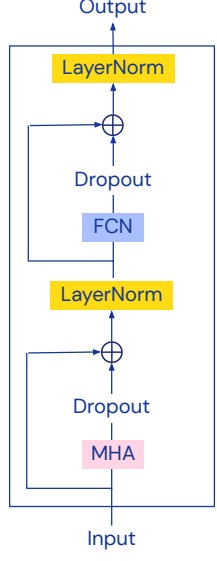

Figure 1: A Transformer block.

## 5 EXPERIMENTAL RESULTS

### 5.1 ASYMPTOTIC TIGHTNESS OF THE UPPER BOUND ON $\mathbf{Lip_\infty(F)}$

A tight bound on the Lipschitz constant of self-attention is desirable for all listed applications in Section 1; it leads to tighter generalisation bounds, lighter constraints for provable robustness, and better expressiveness in residual flow models. Hence we investigate the tightness of our bound on the Lipschitz constant of `L2-MHA`. The Lipschitz constant is a supremum over the space of inputs $X \in \mathbb{R}^{N \times D}$ (c.f. Equation (2)) and approximating it requires solving an intractable optimisation problem. Hence it is infeasible to estimate accurately in general, especially when $X$ is high-dimensional. However, we may compute a lower bound on the Lipschitz

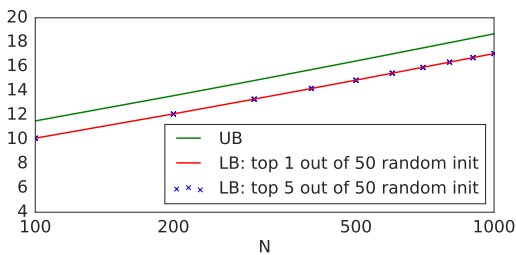

Figure 2: Lower and upper bound on $\mathrm{Lip}_\infty(f)$ for L2-MHA $f$, with $H = D = 1$ and varying $N$.

constant by maximising the norm of the Jacobian $\|J_f(X)\|$ with respect to $X$ until convergence. This local optimum will form a lower bound by Theorem 2.1, and we can expect this lower bound to be fairly tight for the low-dimensional case, provided the optimisation is thorough.

We use this observation to provide empirical evidence for the asymptotic tightness of the upper bound on $\mathrm{Lip}_\infty(f)$ in Theorem 3.2. In Figure 2, we show the upper bound as well as the lower bound on $\mathrm{Lip}_\infty(f)$ obtained by optimising $\|J_f(X)\|_\infty$ with respect to $X$ for `L2-MHA` $f$ with 50 different random initialisations of $X$, with $H = D = 1$ and $N$ varying between 100 and 1000. See Appendix I for further details. Note that we use a log-scale for the x-axis, and recall that the upper bound is $O(\log N - \log \log N)$, dominated by the $O(\log N)$ term for large $N$. Hence the plot for the upper bound shows a linear trend. We also observe that the slope of the lower bound is very similar, providing empirical evidence that the $O(\log N - \log \log N)$ upper bound is asymptotically tight.

There are at least two possible explanations for the gap between the upper and lower bounds. (1) The lower bound is only a local optimum — the true Lipschitz constant is a global optimum across inputs, which can be difficult to attain especially for high values of $N$. (2) The multiplicative constant of the upper bound may be loose. Assuming asymptotic tightness, it remains an open question whether the multiplicative constant can be tightened. We show the analogous plot for $\mathrm{Lip}_2(F)$ and discuss the results in Appendix K. Additionally in Appendix L, we show that optimising $\|J_f(X)\|_\infty$ w.r.t. $X$ for `DP-MHA` $f$ causes the norm to diverge, providing empirical verification of Theorem 3.1, that `DP-MHA` is indeed *not* Lipschitz.

## 5.2 NUMERICAL INVERTIBILITY OF MHA RESIDUAL MAP

Recall from Section 4.1 that $g(x) = x + f(x)$ is invertible if $f$ is contractive. Hence if $f$ is `Contractive-L2-MHA`, $g$ is necessarily invertible. However, technically we do not disprove the invertibility of `DP-MHA`, since the converse does not hold in general i.e. if $f$ is `DP-MHA`, which we have shown is *not* Lipschitz hence *not* contractive, it may still be the case that $g$ *is* invertible. To verify that `DP-MHA` (with the skip connection) is *not* invertible in practice, we compare the numerical invertibility of the residual map $g(x) = x + cf(x)$ between the cases where $f$ is `L2-MHA` and `DP-MHA` in Figure 3. For each, we take MHA with 8 heads and randomly initialised weights, and quantify the

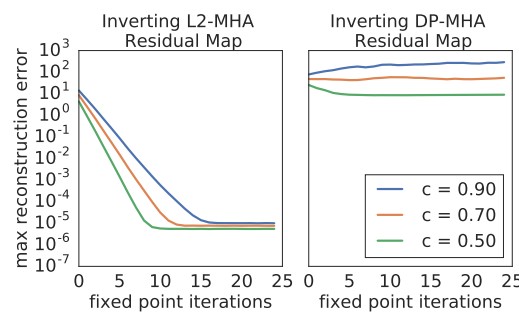

Figure 3: Invertibility of $g(x) = x + cf(x)$ where $f$ is L2-MHA (left) and DP-MHA (right).

maximum reconstruction error across a batch of 128 inputs whose outputs are inverted via the fixed-point iteration described in Section 4.1. We use $N = 64$, $D = 64$, and $c \in \{0.5, 0.7, 0.9\}$ (see Appendix J for analogous results for a wider range of $N$ and $D$ and for `DP-MHA` with trained weights). To highlight the difference between the two types of self-attention, recall in the proof of Theorem 3.1 (showing that `DP-MHA` is not Lipschitz) that when one of the inputs $x_i$ is 0, some terms of the Jacobian grow with the sample variance of $x_{\neq i}$. Hence we check numerical invertibility at a set of $N$ inputs where $x_i = 0$ and $x_{\neq i}$ are chosen uniformly at random. In Figure 3, we see that `DP-MHA` is *not* invertible whereas `L2-MHA` *is* invertible for sufficiently small $c$. This shows how not having the theoretical guarantee of $f$ being contractive can cost us invertibility in practice. We note that the figure shows local invertibility at the sampled inputs, as opposed to global invertibility across the whole input space, yet this clearly highlights the difference between the two choices of self-attention. Experiments with the globally invertible self-attention obtained by normalising with the Lipschitz upper bound are provided in the next section.

## 5.3 EXPRESSIVENESS OF L2-MHA AND INVERTIBLE SELF-ATTENTION

A natural question to ask is: how does the expressiveness of `L2-MHA` and `Contractive-L2-MHA` (that leads to invertible self-attention with the skip connection) compare with the original `DP-MHA`? We expect that the Lipschitz constraint will limit the expressiveness of the Transformer, and would like to find out by how much. We investigate this by comparing the performance of the original Transformer and the Transformer with invertible self-attention (c.f. Figure 1) at character-level language modelling on the Penn Treebank dataset (Marcus et al., 1993). We compare the test negative log-likelihood (NLL) of a baseline LSTM, the original Transformer (`DP-MHA`), and a series of models between the original Transformer and the Transformer with invertible self-attention (`Contractive-L2-MHA`), making one change at a time and tuning the hyperparameters on a validation set. For `Contractive-L2-MHA`, we normalise `L2-MHA` by the bound on $\mathrm{Lip}_\infty(F)$ as it is tighter than the bound on $\mathrm{Lip}_2(F)$. See Appendix I for experimental details.

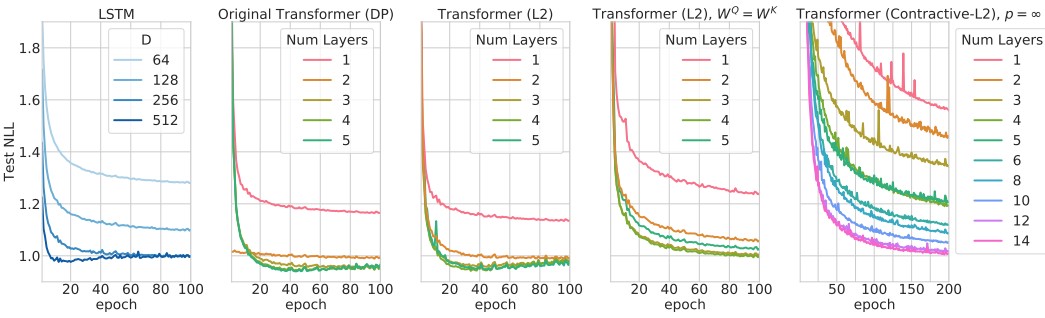

Figure 4: Test NLL curves during training for various LSTM/Transformer models.

The results are shown in Figure 4. The first plot shows the best performing LSTM reaching a test NLL of around 1.0, and the second plot shows the best performing Transformer reaching a slightly improved performance for 3–5 layers of Transformer blocks. We observe instabilities in training for a higher number of layers, requiring careful tuning of the learning rate schedule for stability at the cost of performance, a commonly observed phenomenon in the literature of deep Transformer architectures (Bapna et al., 2018; Parisotto et al., 2020). The third plot shows results for the Transformer with `DP-MHA` replaced with `L2-MHA` but without tying $W^Q$ and $W^K$, and we observe a very similar test performance. The fourth plot shows the change when we further tie the query and key weights (making $W^Q = W^K$); we see that there is a small degradation in performance. Here the number of trainable parameters has been reduced, but in Appendix M we show that matching parameter count does not help performance, suggesting that the reduction in performance when tying queries and keys is not solely due to having fewer parameters. We note that performance saturates at around 5 layers for each Transformer model so far. On the rightmost plot we show results when further dividing self-attention in each block by the upper bound on $\text{Lip}_\infty(F)$, to obtain invertible self-attention. This does give reduced performance for the same number of layers, but we can attain similar performance with more layers, no longer saturating at 5 layers.

Thus we conclude the following. (1) Replacing the dot-product with the L2 distance incurs hardly any loss in expressiveness. (2) Tying the query and key weights to obtain Lipschitz self-attention incurs a small loss in expressiveness. (3) Dividing by the upper bound on $\text{Lip}_\infty(F)$ to obtain invertible self-attention incurs a noticeable loss in expressiveness, but also has a stabilising effect on the optimisation of the Transformer, thus allowing one to compensate for the apparent loss in expressiveness by increasing the number of layers. We show further experimental results that compare the training stability of `DP-MHA` and `(Contractive)-L2-MHA` in Appendix N.

## 6   Conclusion and Discussion

We have shown that the widely used dot-product self-attention is *not* Lipschitz, and that the proposed L2 self-attention *is* Lipschitz, by deriving an $O(\log N - \log \log N)$ Lipschitz bound for $p = \infty$ and an $O(\sqrt{N}(\log N - \log \log N))$ bound for $p = 2$, where $N$ is the input sequence length. We also provided empirical evidence of the asymptotic tightness of the bound for $p = \infty$. Finally we demonstrated that Lipschitz-constrained self-attention can be used to formulate invertible self-attention, which we experimentally evaluated on a character-level language modelling task.

Our approach to Lipschitz self-attention has been to replace the dot-product kernel with an L2 kernel. An alternative would be to constrain the inputs of self-attention to be bounded; if the input space is compact, e.g. $[0, 1]^{N \times D}$, *any* continuously differentiable function is Lipschitz, including dot-product self-attention. However, while being simple to implement, this solution has its own difficulties. First, it makes the Lipschitz constant depend on the range of the input, and thus obtaining a tight bound would require non-trivial mathematical work. We stress that a guarantee that the function is Lipschitz does not tell us anything about its Lipschitz constant; without a tight Lipschitz bound, the true Lipschitz constant can be very large, at which point it is unhelpful that the function is Lipschitz. Second, since self-attention is typically applied at multiple layers within a model (e.g. Transformer), the input to each self-attention will live in a different compact set that depends on the parameters of the previous layers, complicating the analysis for subsequent layers. A solution is to constrain the inputs of each layer to be in the same compact set, e.g. by passing them through a sigmoid non-linearity. This however can have undesirable side effects such as vanishing gradients when the sigmoids are saturated (Hochreiter, 1998). Despite these difficulties, this could be a worthwhile alternative route for obtaining Lipschitz self-attention to explore in the future.

Having a provably Lipschitz self-attention module at our disposal makes it possible to use Transformer-based architectures in applications requiring Lipschitz constraints, while enjoying theoretical guarantees. A natural application of Lipschitz self-attention is for residual flows (Behrmann et al., 2019), and for parameterising Neural ODEs (Chen et al., 2018) where a Lipschitz vector field guarantees the existence of a unique solution to the ODE for all times. These models can be used for density estimation and generative modelling of sets. Another interesting direction for future work would be to analyse different variants of self-attention based on kernels other than dot-product and L2, as Tsai et al. (2019) do from an experimental perspective, for which we believe the mathematical tools developed in this paper may aid the analysis.

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

## A    Chain Rule for vector valued Functions

In this section, we list some useful identities for deriving the Jacobians of the expressions in the paper.

Suppose $\lambda$ is a scalar, $u, v, x$ are column vectors, and $f(u)$ is a vector valued function. We use the standard convention that for $a \in \mathbb{R}^m$, $b \in \mathbb{R}^n$, we have $\frac{\partial a}{\partial b} \in \mathbb{R}^{m \times n}$. Then we have the following chain rule identities:

- $\frac{\partial}{\partial x}[\lambda u] = \lambda \frac{\partial u}{\partial x} + u \frac{\partial \lambda}{\partial x}$
- $\frac{\partial f(u)}{\partial x} = \frac{\partial f(u)}{\partial u} \frac{\partial u}{\partial x}$
- $\frac{\partial}{\partial x}[u^\top v] = u^\top \frac{\partial v}{\partial x} + v^\top \frac{\partial u}{\partial x}$

Note $\frac{\partial \lambda}{\partial x}$ is a row vector, so $u \frac{\partial \lambda}{\partial x}$ is a matrix.

## B    Power Iteration

Although $\|W\|_\infty$ can be computed efficiently in $O(nm)$ time for $W \in \mathbb{R}^{m \times n}$, naïvely computing $\|W\|_2 = \sigma_{\max}(W) := \sqrt{\lambda_{\max}(W^\top W)}$ requires $O(n^3)$ operations. (By $\lambda_{\max}(A)$ we denote the greatest eigenvalue of a symmetric matrix $A$.) We can however obtain an underestimate $\tilde{\sigma}(W)$ via *power iteration*:

$$b_{k+1} = \frac{W^\top W b_k}{\|W^\top W b_k\|_2}, \quad \tilde{\sigma}_k(W) = \sqrt{\frac{b_k^\top W^\top W b_k}{b_k^\top b_k}}, \tag{11}$$

with each iteration taking $O(n^2)$ time. Then using $K \ll n$ iterations gives us an underestimate $\tilde{\sigma}_K$ in $O(Kn^2)$ time. Since this is an underestimate, the resulting approximation to the Lipschitz constant of the linear map will not be an upper bound. However the number of power iterations is usually chosen so that $\tilde{\sigma}$ is accurate enough — $K = 5$ is shown to be sufficient in the context of fully connected networks or convolutions considered by Behrmann et al. (2019).

The iteration will converge if $W^\top W$ has an eigenvalue that is strictly greater in magnitude than its other eigenvalues, and the starting vector $b_0$ has a nonzero component in the direction of an eigenvector associated with the dominant eigenvalue. This happens with probability 1 if $b_0$ is chosen at random, and the convergence is geometric with ratio $|\lambda_2/\lambda_{\max}|$ where $\lambda_2$ is the eigenvalue with second largest magnitude (Mises & Pollaczek-Geiringer, 1929).

## C    Proof of Theorem 3.1 for General $D$

**Theorem 3.1.** DP-MHA is not Lipschitz for any vector $p$-norm $\|\cdot\|_p$ with $p \in [1, \infty]$.

*Proof.* The mapping $f$ can be written as

$$f(X) = PX = \text{softmax}\left(XA^\top X^\top\right) X = \begin{bmatrix} f_1(X)^\top \\ \vdots \\ f_N(X)^\top \end{bmatrix} \in \mathbb{R}^{N \times D}, \tag{12}$$

where $A = W^K W^{Q^\top} / \sqrt{D/H} \in \mathbb{R}^{D \times D}$ and $f_i(X) = \sum_{j=1}^N P_{ij} x_j$ with $P_{i:}^\top = \text{softmax}\left(XAx_i\right)$. Hence $f$ can be interpreted as a map of each $x_i$ to a point in the convex hull of $x_1, ..., x_N$. Since $f$ is a map from $\mathbb{R}^{N \times D}$ to $\mathbb{R}^{N \times D}$, its Jacobian is

$$J_f = \begin{bmatrix} J_{11} & \dots & J_{1N} \\ \vdots & \ddots & \vdots \\ J_{N1} & \dots & J_{NN} \end{bmatrix} \in \mathbb{R}^{ND \times ND}, \tag{13}$$

where $J_{ij} = \frac{\partial f_i(X)}{\partial x_j} \in \mathbb{R}^{D \times D}$. By taking partial derivatives we can show that $J_{ij} = X^\top P^{(i)} \left[ e_{ji} X A^\top + X A \delta_{ij} \right] + P_{ij} I$ where $e_{ij} \in \mathbb{R}^{N \times N}$ is a binary matrix with zeros everywhere

except the $(i,j)$th entry, $\delta_{ij}$ is the Kronecker delta, and $P^{(i)} := \mathrm{diag}(P_{i:}) - P_{i:}^\top P_{i:}$. So for $i = j$:

$$J_{ii} = X^\top P^{(i)} e_{ii} X A^\top + X^\top P^{(i)} X A + P_{ii} I$$
$$= P_{ii}\left(x_i - \sum_k P_{ik} x_k\right) x_i^\top A^\top + X^\top P^{(i)} X A + P_{ii} I. \tag{14}$$

For the last equality, note $e_{ii} X$ has all rows equal to zero except for the $i$th row given by $x_i^\top$. We can then verify that $X^\top P^{(i)} e_{ii} X$ simplifies to $P_{ii}(x_i - \sum_k P_{ik} x_k) x_i^\top$.

For vector $p$-norms, $\|J_f\|_p$ is bounded if and only if its entries are bounded, by definition of the operator norm. The entries of $X^\top P^{(i)} X A$ are bounded for arbitrary $A$ only if the entries of $X^\top P^{(i)} X$ are bounded. So let us investigate the entries of this $D \times D$ matrix. Writing out each term of the matrix, we observe that it is in fact a covariance matrix of a discrete distribution. Specifically:

$$[X^\top P^{(i)} X]_{lm} = \sum_k P_{ik} x_{kl} x_{km} - \left(\sum_k P_{ik} x_{kl}\right)\left(\sum_k P_{ik} x_{km}\right) = \mathrm{Cov}(\mathbb{X}_l, \mathbb{X}_m), \tag{15}$$

where $\mathbb{X}$ is a discrete distribution with support at the inputs $\{x_1, \ldots, x_N\}$ and probability mass function given by their softmax probabilities $\mathbb{P}(\mathbb{X} = x_j) = P_{ij}$. A consequence of this interpretation is that $P^{(i)}$ is *positive semi-definite* (PSD) since for $D = 1$, Equation (15) becomes $X^\top P^{(i)} X = \mathrm{Var}(\mathbb{X}) \geq 0$, with equality if and only if the $x_j$ are all equal.

We use this observation to show that the terms of $J_{ii}$ are unbounded, and so `DP-MHA` is *not* Lipschitz. Consider the case $x_i = 0$. Then $P_{i:}^\top = \mathrm{softmax}(X A x_i) = \frac{1}{N}\mathbb{1}$, i.e. we have uniform attention regardless of $x_{\neq i}$. The first term of $J_{ii}$ in Equation (14) disappears since $x_i = 0$, and the last term becomes $\frac{1}{N} I$. For the second term, the entries $[X^\top P^{(i)} X]_{ll} = \mathrm{Var}(\mathbb{X}_l)$ are unbounded since the latter is equal to the sample variance of $x_{1l}, \ldots, x_{Nl}$, which can be arbitrarily large. $\qquad\square$

## D   Bias term in DP Self-Attention

A natural question to ask is whether we can add bias terms $b^Q$ to $x_i^\top W^Q$ and $b^K$ to $x_j^\top W^K$ to resolve the issue of attention weights $P_{i:}$ becoming uniform when $x_i = 0$. The answer is *no* in general. It can again be shown that $J_{ii}$ is unbounded when $x_i$ is chosen such that $x_i^\top W^Q + b^Q = 0$ (such a choice is possible assuming $W^Q$ is full rank, a dense set in $\mathbb{R}^{D \times D/H}$). Then $P_{i:}^\top = \frac{1}{N}\mathbb{1}$ again, and the diagonal entries of $X^\top P^{(i)} X$ are unbounded.

## E   Efficient Computation of L2 Self-Attention

Dot-product self-attention only requires a few matrix multiplications to compute the logits (i.e. the inputs to the softmax) between all pairs of inputs, without having to loop over pairs, hence it can be computed efficiently. Similarly, we can show that L2 self-attention can also be computed in an efficient manner. Using the identity $\|a - b\|_2^2 = \|a\|_2^2 - 2a^\top b + \|b\|_2^2$ we can compute the logits of L2 attention between all pairs via matrix multiplications and computation of row-wise L2 norms, with negligible overhead compared to dot-product self-attention. Specifically, for L2 self-attention we can show that

$$P = \mathrm{softmax}\left(-\frac{\|XW^Q\|_{\mathrm{row}}^2 \mathbb{1}^\top - 2XW^Q(XW^K)^\top + \mathbb{1}\|XW^K\|_{\mathrm{row}}^{2\top}}{\sqrt{D/H}}\right), \tag{16}$$

where $\|A\|_{\mathrm{row}}^2$ applies the squared L2 norm to each row of $A$, so if $A \in \mathbb{R}^{m \times n}$ then $\|A\|_{\mathrm{row}}^2 \in \mathbb{R}^m$.

In Table 1 we show the wall-clock training times for the Transformer models with different attention functions and a varying number of layers. It is evident that the differences between the models are rather small.

| | 1 Layer | 2 Layers | 3 Layers | 4 Layers | 5 Layers |
|---|---|---|---|---|---|
| Transformer (**DP**) | 37 | 56 | 77 | 92 | 110 |
| Transformer (**L2**) | 35 | 56 | 73 | 99 | 115 |
| Transformer, $W^Q = W^K$ (**L2**) | 39 | 58 | 79 | 91 | 108 |
| Transformer, (**Contractive-L2**) | 37 | 60 | 81 | 102 | 127 |

Table 1: Wall clock training times for one epoch of training (seconds)

## F PROOF OF THEOREM 3.2

Recall the formulation of L2-MHA:

$$F : \mathbb{R}^{N \times D} \to \mathbb{R}^{N \times D}$$

$$F(X) = \left[ f^1(X) W^{V,1}, \ldots, f^H(X) W^{V,H} \right] W^O$$

$$f^h(X) = P^h X A_h$$

$$P_{ij}^h \propto \exp(L_{ij}) := \exp \left( -\frac{\| x_i^\top W^{Q,h} - x_j^\top W^{K,h} \|_2^2}{\sqrt{D/H}} \right), \quad \sum_j P_{ij}^h = 1$$

where we have that $W^{Q,h}, W^{K,h}, W^{V,h} \in \mathbb{R}^{D \times D/H}$, $W^O \in \mathbb{R}^{D \times D}$, $P^h \in \mathbb{R}^{N \times N}$ and $A_h := W^{Q,h} W^{Q,h^\top} / \sqrt{D/H} \in \mathbb{R}^{D \times D}$, and the softmax is applied to each row of the input matrix. Recall Equation (16):

$$P^h = \text{softmax} \left( -\frac{\| XW^{Q,h} \|_{\text{row}}^2 \mathbb{1}^\top - 2XW^{Q,h}(XW^{K,h})^\top + \mathbb{1}\| XW^{K,h} \|_{\text{row}}^{2^\top}}{\sqrt{D/H}} \right).$$

### F.1 L2 SELF-ATTENTION IS *not* LIPSCHITZ FOR GENERAL $W^Q, W^K$

Let us first look at the case of $H = 1$ and suppress the index $h$ to reduce clutter. Consider the map $\tilde{f}(X) := PX$, so $f(X) = \tilde{f}(X)A$. We need $\tilde{f}$ to be Lipschitz for $f$ and hence $F$ to be Lipschitz. Note that $P$ is defined as:

$$P_{ij} \propto \exp(L_{ij}) := \exp \left( -\frac{\| x_i^\top W^Q - x_j^\top W^K \|_2^2}{\sqrt{D/H}} \right)$$

and the normalisation constant satisfies $\sum_j P_{ij} = 1$, for $P \in \mathbb{R}^{N \times N}$, $X \in \mathbb{R}^{N \times D}$.

For L2 self-attention, we may take partial derivatives and use the chain rule to show that the Jacobian of $\tilde{f}$ is:

$$J_{\tilde{f}} = \begin{bmatrix} \tilde{J}_{11} & \ldots & \tilde{J}_{1N} \\ \vdots & \ddots & \vdots \\ \tilde{J}_{N1} & \ldots & \tilde{J}_{NN} \end{bmatrix} \in \mathbb{R}^{ND \times ND} \tag{17}$$

with

$$\tilde{J}_{ij} = X^\top P^{(i)} \frac{\partial L_{i:}}{\partial x_j} + P_{ij} I \in \mathbb{R}^{D \times D} \tag{18}$$

where

$$\frac{\partial L_{i:}}{\partial x_j} = \frac{2}{\sqrt{D/H}} \left[ \left( XW^K - \mathbb{1}x_i^\top W^Q \right) W^{Q^\top} \delta_{ij} + \left( e_{ji} XW^Q - e_{jj} XW^K \right) W^{K^\top} \right] \tag{19}$$

and

$$P^{(i)} := \text{diag}(P_{i:}) - P_{i:}^\top P_{i:} = \begin{bmatrix} P_{i1}(1 - P_{i1}) & -P_{i1}P_{i2} & \ldots & -P_{i1}P_{iN} \\ -P_{i2}P_{i1} & P_{i2}(1 - P_{i2}) & \ldots & -P_{i2}P_{iN} \\ \vdots & \vdots & \ddots & \vdots \\ -P_{iN}P_{i1} & -P_{iN}P_{i2} & \ldots & P_{iN}(1 - P_{iN}) \end{bmatrix},$$

$$P_{ij} = \frac{\exp\left(-\|x_i^\top W^Q - x_j^\top W^K\|_2^2\right)}{\sum_k \exp\left(-\|x_i^\top W^Q - x_k^\top W^K\|_2^2\right)}.$$

Recall that $e_{ji} \in \mathbb{R}^{N\times N}$ is a binary matrix with zeros everywhere except the $(j,i)$th entry. Hence $e_{ji}X$ has all rows equal to zero except for the $j$th row given by $x_i^\top$. We can then verify:

$$X^\top P^{(i)} e_{ji} X = P_{ij}(x_j - \sum_k P_{ik}x_k)x_i^\top. \tag{20}$$

Also note $P^{(i)}$ is symmetric, and each row/colum sums to 0, i.e. $P^{(i)}\mathbb{1} = \mathbb{1}^\top P^{(i)} = 0$. Hence we may simplify the Jacobian terms as follows:

$$\begin{aligned}
\tilde{J}_{ii} &= \frac{2}{\sqrt{D/H}}\left[X^\top P^{(i)}(XW^K - \mathbb{1}x_i^T W^Q)W^{Q^\top} + X^\top P^{(i)}e_{ii}X(W^Q - W^K)W^{K^\top}\right] + P_{ii}I \\
&= \frac{2}{\sqrt{D/H}}\left[X^\top P^{(i)}(XW^K - \mathbb{1}x_i^T W^Q)W^{Q^\top} + P_{ii}(x_i - \sum_k P_{ik}x_k)x_i^\top(W^Q - W^K)W^{K^\top}\right] + P_{ii}I \\
&= \frac{2}{\sqrt{D/H}}\left[X^\top P^{(i)}XW^K W^{Q^\top} + P_{ii}(x_i - \sum_k P_{ik}x_k)x_i^\top(W^Q - W^K)W^{K^\top}\right] + P_{ii}I,
\end{aligned} \tag{21}$$

and for $i \neq j$:

$$\begin{aligned}
\tilde{J}_{ij} &= \frac{2}{\sqrt{D/H}}X^\top P^{(i)}(e_{ij}XW^Q - e_{jj}XW^K)W^{K^\top} + P_{ij}I \\
&= \frac{2}{\sqrt{D/H}}P_{ij}(x_j - \sum_k P_{ik}x_k)(x_i^\top W^Q - x_j^\top W^K)W^{K^\top} + P_{ij}I. \tag{22}
\end{aligned}$$

We are now ready to show that $\tilde{f}$ is *not* Lipschitz for general $W^Q, W^K$:

**Lemma F.1.** *If $W^K \in \mathbb{R}^{D\times D/H}$ is full rank (i.e. full column rank), and $W^K \neq W^Q$, then $J_{ij}$ has terms that are unbounded for $i \neq j$, hence $\tilde{f}$ is not Lipschitz.*

*Proof.* Let us investigate the expression $\tilde{K}_{ij} := P_{ij}W^{K^\top}(x_j - \sum_k P_{ik}x_k)(x_i^\top W^Q - x_j^\top W^K) \in \mathbb{R}^{\frac{D}{H}\times\frac{D}{H}}$ for $i \neq j$, which is related to $\tilde{J}_{ij}$ as follows by Equation (22):

$$W^{K^\top}\tilde{J}_{ij} = \left(\frac{2}{\sqrt{D/H}}\tilde{K}_{ij} + P_{ij}I\right)W^{K^\top}.$$

It suffices to show that $\tilde{K}_{ij}$ is unbounded to show that $\tilde{J}_{ij}$ is unbounded, since $W^K$ is full rank and $P_{ij} \in [0,1]$.

Let $y_j^\top = x_i^\top W^Q - x_j^\top W^K$. Then we have:

$$\begin{aligned}
y_j - \sum_k P_{ik}y_k &= W^{Q^\top}x_i - W^{K^\top}x_j - \sum_k P_{ik}(W^{Q^\top}x_i - W^{K^\top}x_k) \\
&= W^{Q^\top}x_i - W^{K^\top}x_j - (W^{Q^\top}x_i - \sum_k P_{ik}W^{K^\top}x_k) \\
&= -W^{K^\top}(x_j - \sum_k P_{ik}x_k).
\end{aligned}$$

Hence $\tilde{K}_{ij} = -P_{ij}(y_j - \sum_k P_{ik}y_k)y_j^\top$. Note $y_i$ can take an arbitrary value in $\mathbb{R}^{D/H}$, since $W^K \neq W^Q$ and $W^K$ is full-rank.

For all $j \neq i$, let us choose $x_j$ such that $y_j = -y_i$. This is possible for any value of $y_i$ since $W^K$ is full-rank. Note $y_j = -y_i$ and not $y_i$. We then have that $\|y_j\|_2^2$ is equal for all $j$, hence

$P_{ij} := \frac{\exp(-\|y_j\|_2^2)}{\sum_k \exp(-\|y_k\|_2^2)} = \frac{1}{N}$ for all $j$. Then for $i \neq j$, $\tilde{K}_{ij}$ simplifies to

$$\tilde{K}_{ij} = -\frac{1}{N}\left(-y_i - \frac{1}{N}(N-2)(-y_i)\right)(-y_i)^\top = -\frac{2N-2}{N^2}y_iy_i^\top$$

whose entries are unbounded since $y_i$ can be any vector in $\mathbb{R}^{D/H}$ (note we assume $N \geq 2$ for self-attention to be well-defined, hence $2N - 2 \neq 0$). $\qquad\square$

The intuition for this result is as follows: a reason for DP-MHA not being Lipschitz is that for $x_i = 0,$, the attention weights $P_{ij}$ become uniform regardless of the values of $x_j$ for $j \neq i$. A similar issue arises for L2-MHA with $W^Q \neq W^K$ and full-rank $W^K$, as shown above: given any $x_i$, we can choose $x_j$ such that the $P_{ij}$ become uniform.

## F.2  L2 SELF-ATTENTION IS LIPSCHITZ FOR $W^Q = W^K$

Hence we impose the restriction that $W^K = W^Q$. With this assumption we have

$$P_{ij} \propto \exp\left(-\|(x_i - x_j)^\top\sqrt{A}\|_2^2\right) \tag{23}$$

where $A = W^QW^{Q^\top}/\sqrt{D/H} \in \mathbb{R}^{D\times D}$ and $\sqrt{A}$ is chosen such that $A = \sqrt{A}\sqrt{A}^\top$, in particular $\sqrt{A} := W^Q/(D/H)^{\frac{1}{4}}$. The terms in the Jacobian of $\tilde{f}$ simplify to:

$$\tilde{J}_{ii} = 2X^\top P^{(i)}XA + P_{ii}I \quad (\text{note } P^{(i)}\mathbb{1} = 0), \tag{24}$$

$$\tilde{J}_{ij} = 2P_{ij}(x_j - \sum_k P_{ik}x_k)(x_i - x_j)^\top A + P_{ij}I \quad \text{for } i \neq j. \tag{25}$$

Let the Jacobian of $f(X)$ be:

$$J_f = \begin{bmatrix} J_{11} & \dots & J_{1N} \\ \vdots & \ddots & \vdots \\ J_{N1} & \dots & J_{NN} \end{bmatrix} \in \mathbb{R}^{ND\times ND}. \tag{26}$$

Since $f(X) = \tilde{f}(X)A$, and by the chain rule $\frac{\partial}{\partial x_j}[\tilde{f}_i(X)A] = A^\top\frac{\partial\tilde{f}_i(X)}{\partial x_j} = A\frac{\partial\tilde{f}_i(X)}{\partial x_j}$ (by symmetry of $A$), we have that $J_{ij} = A\tilde{J}_{ij}$. Hence

$$J_{ii} = 2AX^\top P^{(i)}XA + P_{ii}A \quad (\text{note } P^{(i)}\mathbb{1} = 0), \tag{27}$$

$$J_{ij} = 2P_{ij}A(x_j - \sum_k P_{ik}x_k)(x_i - x_j)^\top A + P_{ij}A \quad \text{for } i \neq j. \tag{28}$$

Noting $\mathrm{Lip}_p(f) = \sup_X\|J_f(X)\|_p$, we would like to upper bound $\|J_f\|_p$.

### F.2.1  UPPER BOUND ON $\mathrm{Lip}_\infty(F)$ FOR L2-MHA

Consider the choice $p = \infty$, where $\|J_f\|_\infty$ is the maximum absolute row sum of $J_f$. A key observation is that if we can bound the $\infty$-norm of the Jacobian of $f_i$, a single output of $f$ (i.e. a single block row $\|[J_{i1}, ..., J_{iN}]\|_\infty$ of $J_f$), then this is also a bound on $\|J_f\|_\infty$ due to permutation equivariance of self-attention; all block rows have the same maximal $\|\cdot\|_\infty$ when each is optimised over the input $X$. Using this, we can prove that $\|J_f\|_\infty$ admits an upper bound that is $O(\log N - \log\log N)$. Below we state and prove lemmas that lead to the proof of this upper bound.

First we analyse the term $\sqrt{A}^\top X^\top P^{(i)}X\sqrt{A}$, that appears in the first term of $J_{ii}$. Note that for $Y := X\sqrt{A}$, so that the rows of $Y$ are $y_i^\top := x_i^\top\sqrt{A}$, we have

$$\sqrt{A}^\top X^\top P^{(i)}X\sqrt{A} = Y^\top P^{(i)}Y = \mathrm{Cov}(\mathbb{Y}) \tag{29}$$

where $\mathbb{P}(\mathbb{Y} = y_j) = P_{ij} = \exp(-\|y_j - y_i\|_2^2)/\sum_k\exp(-\|y_k - y_i\|_2^2)$. The last equality uses the observation in Equation (9).

The central inequality used throughout the proof of the main theorem is the following:

**Lemma F.2.** $\mathrm{Tr}(\mathrm{Cov}(\mathbb{Y})) = \sum_j P_{ij} \|y_j - \sum_k P_{ik} y_k\|_2^2 \le \sum_j P_{ij} \|y_j - y_i\|_2^2 \le \phi^{-1}(N-1)$ *where* $\phi(c) = c \exp(c+1)$ *is a one-dimensional invertible function on* $\mathbb{R}_{\ge 0}$.

*Proof.* The first equality holds since $\mathrm{Tr}(\mathrm{Cov}(\mathbb{Y})) = \sum_j \mathrm{Cov}(\mathbb{Y})_{jj} = \sum_j \mathrm{Var}(\mathbb{Y}_j) = \sum_j \mathbb{E}[(\mathbb{Y}_j - \mathbb{E}[\mathbb{Y}_j])^2]$. The next inequality holds since $\mathrm{Var}(\mathbb{Y}_j) = \mathrm{Var}(\overline{\mathbb{Y}}_j) = \mathbb{E}[\overline{\mathbb{Y}}_j^2] - \mathbb{E}[\overline{\mathbb{Y}}_j]^2 \le \mathbb{E}[\overline{\mathbb{Y}}_j^2]$ where $\overline{\mathbb{Y}} = \mathbb{Y} - y_i$. The final inequality can be proved as follows.

We would like to bound

$$\sum_j P_{ij} \|y_j - y_i\|_2^2 = \frac{\sum_j \|y_j - y_i\|_2^2 \exp(-\|y_j - y_i\|_2^2)}{\sum_k \exp(-\|y_k - y_i\|_2^2)} = \frac{\sum_j z_j^2 \exp(-z_j^2)}{\sum_k \exp(-z_k^2)} \tag{30}$$

where $z_j := \|y_j - y_i\|_2$ (hence $z_i = 0$). Define:

$$g(z) := \frac{\sum_j z_j^2 \exp(-z_j^2)}{\sum_k \exp(-z_k^2)} = \frac{\sum_{j \neq i} z_j^2 \exp(-z_j^2)}{1 + \sum_{k \neq i} \exp(-z_k^2)}. \tag{31}$$

First note that as $z_j \to \infty$, $\exp(-z_j^2) \to 0$ exponentially fast, causing the product $z_j^2 \exp(-z_j^2) \to 0$. Hence we expect the above quantity to be bounded and attain its maximum.

Let $h(z_j) := \exp(-z_j^2)$ for notational conciseness, and note $h(z_j) > 0$. By taking partial derivatives with the chain rule, we have that for $j \neq i$

$$\frac{\partial g(z)}{\partial z_j} = \frac{2 y_j h(z_j)}{(\sum_k h(z_k))^2} \left[ (1 - z_j^2) \sum_k h(z_k) + \sum_k h(z_k) z_k^2 \right]. \tag{32}$$

Hence the derivative is 0 if and only if $z_j = 0$ or $(1 - z_j^2) \sum_k h(z_k) + \sum_k h(z_k) z_k^2 = 0$, the latter being equivalent to $z_j^2 = 1 + \frac{\sum_k h(z_k) z_k^2}{\sum_k h(z_k)} = 1 + g(z)$. Hence at the maximum, the non-zero values among $\{z_j\}_{j=1}^N$ must be equal to one another. It is clear now that the maximum value $c$ is attained when $z_j^2 = 1 + c$ for $j \neq i$ (and recall $z_i = 0$). So $h(z_j) = \exp(-1 - c)$ for $j \neq i$. Substituting this into $g(z)$, and rearranging, we obtain $c \exp(c + 1) = N - 1$. Note $\phi(x) := x \exp(x + 1)$ is increasing for $x > 0$ hence $c = \phi^{-1}(N - 1)$. $\qquad\square$

Note $\phi(\log N) = (\log N) \exp(\log N + 1) \ge N \log N \ge N - 1$ for $N \ge 3$. Since $\phi$ is increasing, we have $\phi^{-1}(N-1) \le \log(N)$ for $N \ge 3$. In fact, it is known that $\phi^{-1}(N-1) = O(\log N - \log \log N)$ (Corless et al., 1996).

Note the $A$ term in $f(X) = \tilde{f}(X) A$ allows us to use the above inequality, since $Y^\top P^{(i)} Y = \mathrm{Cov}(\mathbb{Y})$ now appears in the terms of $J_f$:

$$J_{ii} = 2\sqrt{A}[Y^\top P^{(i)} Y]\sqrt{A}^\top + P_{ii} A, \tag{33}$$

$$J_{ij}, = 2\sqrt{A} P_{ij}(y_j - \sum_k P_{ik} y_k)(y_i - y_j)^\top \sqrt{A}^\top + P_{ij} A \ \text{ for } i \neq j. \tag{34}$$

Using the inequalities $\|BC\| \le \|B\|\|C\|$, $\|B + C\| \le \|B\| + \|C\|$ and $\|[A_1, \ldots, A_N]\| \le \sum_i \|A_i\|$, we have:

$\|[J_{i1}, \ldots, J_{iN}]\|_\infty$

$\le \|J_{ii}\|_\infty + \sum_{j \neq i} \|J_{ij}\|_\infty$

$\le 2\|\sqrt{A}\|_\infty \|Y^\top P^{(i)} Y\|_\infty \|\sqrt{A}^\top\|_\infty + P_{ii} \|A\|_\infty$

$\quad + 2 \sum_{j \neq i} \|\sqrt{A}\|_\infty \|P_{ij}(y_j - \sum_k P_{ik} y_k)(y_i - y_j)^\top\|_\infty \|\sqrt{A}^\top\|_\infty + P_{ij} \|A\|_\infty$

$= 2\|\sqrt{A}\|_\infty \|\sqrt{A}^\top\|_\infty \left( \|Y^\top P^{(i)} Y\|_\infty + \sum_{j \neq i} \|P_{ij}(y_j - \sum_k P_{ik} y_k)(y_i - y_j)^\top\|_\infty \right) + \|A\|_\infty$

$= 2 \frac{\|W^Q\|_\infty \|W^{Q^\top}\|_\infty}{\sqrt{D/H}} \left( \|Y^\top P^{(i)} Y\|_\infty + \sum_j \|P_{ij}(y_j - \sum_k P_{ik} y_k)(y_i - y_j)^\top\|_\infty \right) + \frac{\|W^Q W^{Q^\top}\|_\infty}{\sqrt{D/H}}.$

For the first equality, note that $\sum_j P_{ij} = 1$. For the second equality, note that the summand for $j = i$ is 0 because the term $y_i - y_j = 0$. Each of the terms in the brackets are bounded by the following lemmas:

**Lemma F.3.** $\|Y^\top P^{(i)} Y\|_\infty \le \phi^{-1}(N-1)\sqrt{D/H}$ ($\phi$ defined as in Lemma F.2).

*Proof.* Recall that $Y^\top P^{(i)} Y = \mathrm{Cov}(\mathbb{Y})$. Let $\sigma(\mathbb{Y}_m)$ denote the standard deviation of $\mathbb{Y}_m$. Then $[\mathrm{Cov}(\mathbb{Y})]_{lm} \le \sigma(\mathbb{Y}_l)\sigma(\mathbb{Y}_m)$. Hence

$$\|\mathrm{Cov}(\mathbb{Y})\|_\infty = \max_l \sum_m |[\mathrm{Cov}(\mathbb{Y})]_{lm}| \le \max_l \sigma(\mathbb{Y}_l) \sum_m \sigma(\mathbb{Y}_m)$$

$$\le \sqrt{\frac{D}{H}} \sum_m \sigma^2(\mathbb{Y}_m) = \sqrt{\frac{D}{H}} \mathrm{Tr}(\mathrm{Cov}(\mathbb{Y}))$$

$$\le \sqrt{\frac{D}{H}} \phi^{-1}(N-1),$$

since $\sum_m \sigma(\mathbb{Y}_m) \le \sqrt{\frac{D}{H}}\sqrt{\sum_m \sigma^2(\mathbb{Y}_m)}$ (by e.g. using the Cauchy–Schwartz inequality on $[\sigma(\mathbb{Y}_1), \ldots, \sigma(\mathbb{Y}_{D/H})]$ and $\mathbb{1}$) and $\max_l \sigma(\mathbb{Y}_l) \le \sqrt{\sum_m \sigma^2(\mathbb{Y}_m)}$, and the last inequality is from Lemma F.2. $\qquad\square$

**Lemma F.4.** $\sum_j \|P_{ij}(y_j - \sum_k P_{ik}y_k)(y_i - y_j)^\top\|_\infty \le \phi^{-1}(N-1)\sqrt{D/H}$.

*Proof.* Note $\|ab^\top\|_\infty = \|a\|_\infty\|b\|_1$ for real vectors $a, b$. Hence

$$\sum_j \|P_{ij}(y_j - \sum_k P_{ik}y_k)(y_i - y_j)^\top\|_\infty = \sum_j P_{ij}\|y_j - \sum_k P_{ik}y_k\|_\infty\|y_i - y_j\|_1$$

$$= a^\top b \le \|a\|_2\|b\|_2,$$

where $a_j = \sqrt{P_{ij}}\|y_j - \sum_k P_{ik}y_k\|_\infty$, $b_j = \sqrt{P_{ij}}\|y_i - y_j\|_1$.

Note $a_j \le c_j := \sqrt{P_{ij}}\|y_j - \sum_k P_{ik}y_k\|_2$ since $\|x\|_\infty \le \|x\|_2$ for vector $x$. Hence $\|a\|_2 \le \|c\|_2$. Also $b_j \le \sqrt{\frac{D}{H}}d_j := \sqrt{\frac{D}{H}}\sqrt{P_{ij}}\|y_i - y_j\|_1$ since $\|x\|_1 \le \sqrt{\frac{D}{H}}\|x\|_2$ (e.g. by the Cauchy–Schwartz inequality on $[|x_1|, \ldots, |x_{D/H}|]$ and $\mathbb{1}$) for $x \in \mathbb{R}^{D/H}$. Hence $\|b\|_2 \le \sqrt{\frac{D}{H}}\|d\|_2$.

Note $\|c\|_2^2 = \sum_j P_{ij}\|y_j - \sum_k P_{ik}y_k\|_2^2 = \mathrm{Tr}(\mathrm{Cov}(\mathbb{Y})) \le \phi^{-1}(N-1)$ from Lemma F.2, and $\|d\|_2^2 = \sum_j P_{ij}\|y_i - y_j\|_2^2 \le \phi^{-1}(N-1)$ also from Lemma F.2. Hence $\|a\|_2\|b\|_2 \le \sqrt{\frac{D}{H}}\|c\|_2\|d\|_2 \le \sqrt{\frac{D}{H}}\phi^{-1}(N-1)$. $\qquad\square$

Putting the above lemmas altogether, with the observation $\sup_X \|J_f(X)\|_\infty = \sup_X \|[J_{i1}(X), \ldots, J_{iN}(X)]\|_\infty$ by permutation invariance of $\|J_f\|_\infty$ (since $f$ is permutation equivariant and $\|\cdot\|_\infty$ is the maximum absolute row sum), we have

$$\|J_f\|_\infty \le 4\|W^Q\|_\infty\|W^{Q^\top}\|_\infty\phi^{-1}(N-1) + \frac{\|W^Q W^{Q^\top}\|_\infty}{\sqrt{D/H}}$$

$$\le \|W^Q\|_\infty\|W^{Q^\top}\|_\infty\left(4\phi^{-1}(N-1) + \frac{1}{\sqrt{D/H}}\right) \qquad (35)$$

$$\le \|W^Q\|_\infty\|W^{Q^\top}\|_\infty\left(4\log N + \frac{1}{\sqrt{D/H}}\right),$$

where the last inequality holds for $N \ge 3$.

The full multihead attention map that combines the heads $f^h(X)$ is:

$$F : X \mapsto \left[f^1(X)W^{V,1}, \ldots f^H(X)W^{V,H}\right]W^O = g(X)W^V W^O$$

where $g : X \mapsto [f^1(X), \ldots, f^H(X)]$, $W^O \in \mathbb{R}^{D \times D}$ and

$$W^V = \begin{bmatrix} W^{V,1} & \cdots & 0 \\ \vdots & \ddots & \vdots \\ 0 & \cdots & W^{V,H} \end{bmatrix} \in \mathbb{R}^{DH \times D}.$$

Note the Jacobian $J_g$ is a block matrix whose rows are $J_{f^h}$, hence $\|J_g\|_\infty = \max_h \|J_{f^h}\|_\infty$, and similarly $\|W^{V^\top}\|_\infty = \max_h \|W^{V,h^\top}\|_\infty$. Hence we have

$$\mathrm{Lip}_\infty(F) \le \max_h \|J_{f^h}\|_\infty \max_h \|W^{V,h^\top}\|_\infty \|W^{O^\top}\|_\infty.$$

Combining this with Inequality (35), we have:

$$\mathrm{Lip}_\infty(F) \le \left( 4\phi^{-1}(N-1) + \frac{1}{\sqrt{D/H}} \right) \max_h \|W^{Q,h}\|_\infty \|W^{Q,h^\top}\|_\infty \max_h \|W^{V,h^\top}\|_\infty \|W^{O^\top}\|_\infty.$$

### F.2.2 Upper bound on $\mathbf{Lip_2}(F)$ for L2-MHA

For $p = 2$, we use the following lemma:

**Lemma F.5.** *Let $A$ be a block matrix with block rows $A_1, \ldots, A_N$. Then $\|A\|_2 \le \sqrt{\sum_i \|A_i\|_2^2}$, and equality holds if and only if the first right singular vectors of the $A_i$ align.*

*Proof.*

$$\|A\|_2^2 = \left\| \begin{bmatrix} A_1 \\ \vdots \\ A_N \end{bmatrix} \right\|_2^2 = \sup_{\|x\|_2 = 1} \left\| \begin{bmatrix} A_1 \\ \vdots \\ A_N \end{bmatrix} x \right\|_2^2 = \sup_{\|x\|_2 = 1} \sum_i \|A_i x\|_2^2 \le \sum_i \sup_{\|x\|_2 = 1} \|A_i x\|_2^2 = \sum_i \|A_i\|_2^2.$$

Note that equality holds if and only if the first right singular vectors of the $A_i$ align. $\qquad\square$

Hence a bound on the spectral norm of each block row of $J_f$ can give us an $O(\sqrt{N})$ bound on $\|J_f\|_2$, which may be loose, and it remains an open question as to whether this bound can be tightened.

To bound the $\|\cdot\|_2$ norm of each row of $J_f$, we use the following lemmas:

**Lemma F.6.** $\|Y^\top P^{(i)} Y\|_2 \le \phi^{-1}(N-1)$

*Proof.* $\|Y^\top P^{(i)} Y\|_2 = \|\mathrm{Cov}(\mathbb{Y})\|_2 = \lambda_{\max}(\mathrm{Cov}(\mathbb{Y})) \le \mathrm{Tr}(\mathrm{Cov}(\mathbb{Y})) \le \phi^{-1}(N-1)$, where the first equality holds by symmetry of $\mathrm{Cov}(\mathbb{Y})$ and the next holds by $\mathrm{Cov}(\mathbb{Y})$ being positive semi-definite, so all its eigenvalues are non-negative, and hence the maximal eigenvalue is bounded by the sum of the eigenvalues, equal to its trace. The final inequality is from Lemma F.2. $\qquad\square$

**Lemma F.7.** $\sum_j \|P_{ij}(y_j - \sum_k P_{ik} y_k)(y_i - y_j)^\top\|_2 \le \phi^{-1}(N-1)$

*Proof.* Directly use Cauchy–Schwartz on $c$ and $d$ in the proof of Lemma F.4. $\qquad\square$

Again using the inequalities $\|BC\| \le \|B\|\|C\|$, $\|B + C\| \le \|B\| + \|C\|$ and $\|[A_1, \ldots, A_N]\| \le \sum_i \|A_i\|$, with the additional equality $\|B^\top\|_2 = \|B\|_2$, we have the bound:

$$\|[J_{i1}, \ldots, J_{iN}]\|_2$$

$$\le 2 \frac{\|W^Q\|_2 \|W^{Q^\top}\|_2}{\sqrt{D/H}} \left( \|Y^\top P^{(i)} Y\|_2 + \sum_j \|P_{ij}(y_j - \sum_k P_{ik} y_k)(y_i - y_j)^\top\|_2 \right) + \frac{\|W^Q W^{Q^\top}\|_2}{\sqrt{D/H}}$$

$$\le 4\phi^{-1}(N-1) \frac{\|W^Q\|_2^2}{\sqrt{D/H}} + \frac{\|W^Q W^{Q^\top}\|_2}{\sqrt{D/H}}$$

$$\le \frac{\|W^Q\|_2^2}{\sqrt{D/H}} \left( 4\phi^{-1}(N-1) + 1 \right).$$

Using Lemma F.5, we have that

$$\|J_f\|_2 \leq \frac{\sqrt{N}\|W^Q\|_2^2}{\sqrt{D/H}}\left(4\phi^{-1}(N-1)+1\right) \tag{36}$$

$$\leq \frac{\sqrt{N}\|W^Q\|_2^2}{\sqrt{D/H}}(4\log N + 1).$$

To obtain the final result for the full multihead self-attention $F$, we need a final lemma:

**Lemma F.8.** *Let $A$ be a block matrix with block columns $A_1, \ldots, A_N$. Then $\|A\|_2 \leq \sqrt{\sum_i \|A_i\|_2^2}$.*

*Proof.*

$$\|A\|_2 = \|[A_1, \ldots, A_N]\|_2 = \sup_{\sum_i \|x_i\|_2^2 = 1}\left\|[A_1, \ldots, A_N]\begin{bmatrix} x_1 \\ \vdots \\ x_N \end{bmatrix}\right\|_2^2 = \sup_{\sum_i \|x_i\|_2^2 = 1}\|\sum_i A_i x_i\|_2$$

$$\leq \sup_{\sum_i \|x_i\|_2^2 = 1}\sum_i \|A_i x_i\|_2 = \sup_{\|e_i\|_2 = 1, \sum_i \lambda_i^2 = 1}\sum_i \lambda_i \|A_i e_i\|_2 = \sup_{\sum_i \lambda_i^2 = 1}\sum_i \lambda_i \|A_i\|_2$$

$$\leq \sqrt{\sum_i \|A_i\|_2^2},$$

where we are using the substitution $x_i = \lambda_i e_i$, and the last inequality holds by e.g. Cauchy–Schwartz inequality on $[\lambda_1, \ldots, \lambda_N]$ and $[\|A_1\|_2, \ldots, \|A_N\|_2]$. $\square$

Recall that

$$F : X \mapsto \left[f^1(X)W^{V,1}, \ldots, f^H(X)W^{V,H}\right]W^O.$$

Since $\|f^h(X)W^{V,h}\|_2 \leq \|J_{f^h}\|_2\|W^{V,h}\|_2$, by Lemma F.8 we have that

$$\left\|[f^1(X)W^{V,1}, \ldots, f^H(X)W^{V,H}]\right\|_2 \leq \sqrt{\sum_h \|J_{f^h}\|_2^2\|W^{V,h}\|_2^2}$$

and hence

$$\mathrm{Lip}_2(F) \leq \left(\sqrt{\sum_h \|J_{f^h}\|_2^2\|W^{V,h}\|_2^2}\right)\|W^O\|_2. \tag{37}$$

Combining this with Inequality (36), we have:

$$\mathrm{Lip}_2(F) \leq \frac{\sqrt{N}}{\sqrt{D/H}}\left(4\phi^{-1}(N-1)+1\right)\left(\sqrt{\sum_h \|W^{Q,h}\|_2^2\|W^{V,h}\|_2^2}\right)\|W^O\|_2.$$

## G  THE CASE WITH MASKING

Since self-attention is often used with *masking*, a natural question is how masking affects the derived bounds. In self-attention (for any choice of attention function), masking is implemented as follows: given a set of mask indices $\mathcal{M} \subset \{1, \ldots, N\} \times \{1, \ldots, N\}$, the logits (i.e. the inputs to the softmax) are set to $-\infty$ at the mask indices. That is,

$$L_{ij} = \begin{cases} \tilde{L}_{ij} & \text{if } (i,j) \notin \mathcal{M} \\ -\infty & \text{if } (i,j) \in \mathcal{M} \end{cases}$$

where $\tilde{L}_{ij}$ is the original logit (e.g. for L2 self-attention, $\tilde{L}_{ij} = -(x_i - x_j)^\top A(x_i - x_j)$).

Masking implies $f_i(X)$ is not a function of $x_j$ for $(i,j) \in \mathcal{M}$, hence $J_{ij} = 0$ for $(i,j) \in \mathcal{M}$. Thus $f_i(X)$ is equal to the $i$th output for self-attention with inputs restricted to $\{x_j : (i,j) \notin \mathcal{M}\}$, the unmasked inputs with respect to the $i$th output. Hence $J_{ij}$ will no longer contribute to the bound on $\|[J_{i1}, \ldots, J_{iN}]\|$, and hence the bound for the unmasked case will continue to hold as long as $(i,i) \in \mathcal{M}$ i.e. $x_i$ attends to itself (this is necessary for the proof of Lemma F.2 to hold). The bound can in fact be tightened by replacing $N$ with $|\{x_j : (i,j) \notin \mathcal{M}\}|$, the number of unmasked inputs with respect to the $i$th output.

## H    DROPOUT IS CONTRACTIVE

At test time, `Dropout` multiplies inputs by the dropout keep probability $p < 1$, so it is a contraction with Lipschitz constant $p$ at evaluation time. At training time, `Dropout` amounts to setting some inputs to zero, while keeping other inputs constant. This can be expressed as right multiplication by a diagonal binary matrix $M$, and for such matrices we can verify $\|M\|_p := \sup_{\|x\|_p=1} \|Mx\|_p \leq 1$.

## I    EXPERIMENTAL DETAILS

For the experiment in Section 5.1, showing the asymptotic tightness of the upper bound on $\text{Lip}_\infty(F)$ where $F$ is `L2-MHA`, we fix all free parameters of $F$ (namely $W^Q, W^V$) to be the identity, and only optimise the input $X$. We use 50 random initialisations of $X$ for each $N$, where $X_{ij} \sim \mathcal{U}[-c, c]$ for $c \sim \mathcal{U}[0, 10]$ (we observed that having $c$ itself be random improves optimisation). We display the top 5 results for each value of $N$ after optimising each random initialisation till convergence using Adam (Kingma & Ba, 2015) with a learning rate of $0.1$.

For the experiments in Section 5.3, we comparing the performance of the original Transformer and the Transformer with Lipschitz/invertible self-attention at character-level language modelling on the Penn Treebank dataset (Marcus et al., 1993).[1] Each training example is a sentence represented as a variable-length sequence of characters, and examples are batched according to length such that padding is minimised, with the maximum sequence length set to 288. All models are autoregressive, outputting the logits for the categorical likelihood predicting the next character, and are trained using maximum likelihood (cross-entropy loss) with a batch size of 64. The LSTM models have the dimensionality of the hidden state equal to the dimensionality $D$ of the cell state (the usual default implementation). The Transformer models are trained with a varying number of blocks (number of layers) with $H = 8$ heads and $D = 512$, tuning hyperparameters for dropout rate in $\{0, 0.1, 0.2\}$ and base learning rate $\gamma \in \{0.2, 0.4, 0.6, 0.8, 1.0, 1.5, 2.0\}$ with number of warmup iterations $w \in \{1000, 2000, 4000, 8000\}$ for the standard custom learning rate schedule in Vaswani et al. (2017):

$$\epsilon_t = \frac{\gamma}{\sqrt{D}} \min(t^{-1/2}, tw^{-3/2}),$$

where $\epsilon_t$ is the learning rate at training iteration $t$. Hence the learning rate linearly increases from $0$ to $(Dw)^{-1/2}$ over $w$ iterations, then decays proportionally to $t^{-1/2}$. We use Glorot Uniform initialisation (Glorot & Bengio, 2010) for all weights ($U\left[-\sqrt{\frac{1}{d_{in}+d_{out}}}, \sqrt{\frac{1}{d_{in}+d_{out}}}\right]$), except for weights in `L2-MHA` that are initialised from $U\left[-\frac{s}{\sqrt{D}}, \frac{s}{\sqrt{D}}\right]$, and $s$ is a hyperparameter. For $D = 512$, we used $s = \frac{1}{2^4}$. All experiments were done in Tensorflow 1.14 (Abadi et al., 2016). The code will be released upon de-anonymisation.

In Table 2 we show the best Test NLL across training of Transformer models in Figure 4.

| Number of Layers | 1 | 2 | 3 | 4 | 5 | 6 | 8 | 10 | 12 | 14 |
|---|---|---|---|---|---|---|---|---|---|---|
| **(DP)** | 1.164 | 0.989 | 0.954 | 0.943 | 0.940 | - | - | - | - | - |
| **(L2)** | 1.134 | 0.987 | 0.958 | 0.942 | 0.946 | - | - | - | - | - |
| $W^Q = W^K$ **(L2)** | 1.237 | 1.055 | 1.001 | 0.994 | 1.023 | - | - | - | - | - |
| **(Contractive-L2)** | 1.558 | 1.453 | 1.343 | 1.191 | 1.205 | 1.119 | 1.086 | 1.0495 | 1.015 | 1.005 |

Table 2: Test NLL for Transformer models on PTB character level language modelling

## J    NUMERICAL INVERTIBILITY OF MHA RESIDUAL MAP

Following Section 5.2, Figure 5 confirms that numerical invertibility does not hold for trained weights for dot-product multihead self-attention (DP-MHA) (obtained from one-layer Transformer (DP)

---

[1]We use the standard training-validation-test split, and the dataset can be found at e.g. `https://github.com/harvardnlp/TextFlow/tree/master/data/ptb`.

model used for Figure 4), similar to the randomly initialised weight case. Figure 6 shows additional results for different values of $N$ and $D$.

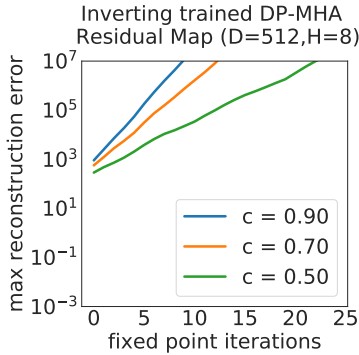

Figure 5: Invertibility of $g(x) = x + cf(x)$ for trained DP-MHA $f$.

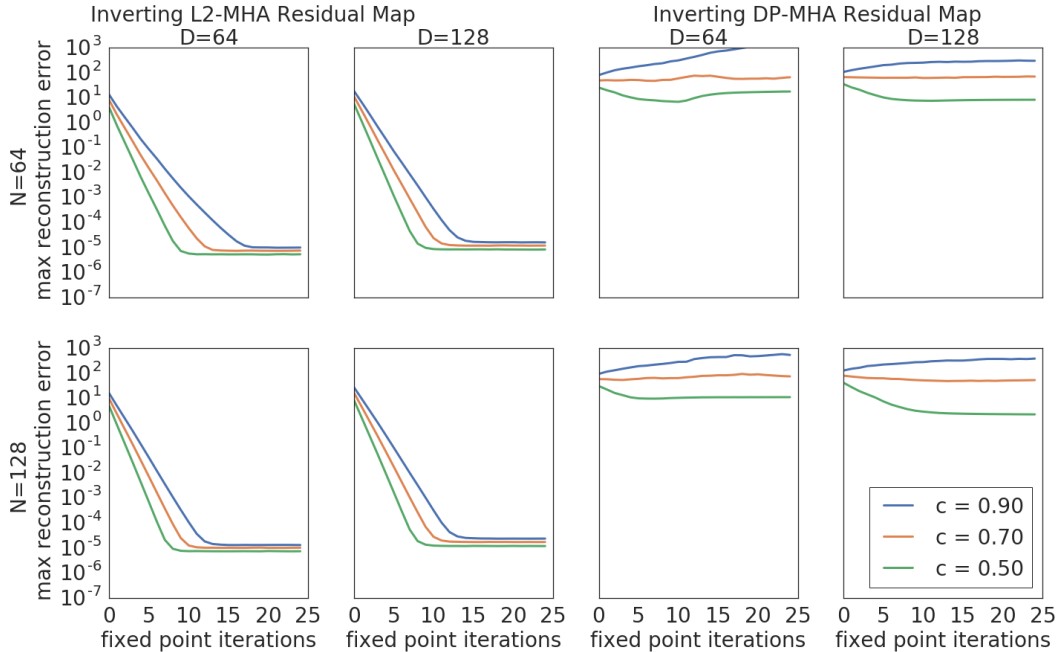

Figure 6: Numerical invertibility of $g(x) = x + cf(x)$ where $f$ is L2-MHA(left) or DP-MHA (right), for different values of $N$ and $D$.

## K  BEHAVIOUR OF LOWER BOUND ON $\mathrm{Lip}_2(F)$

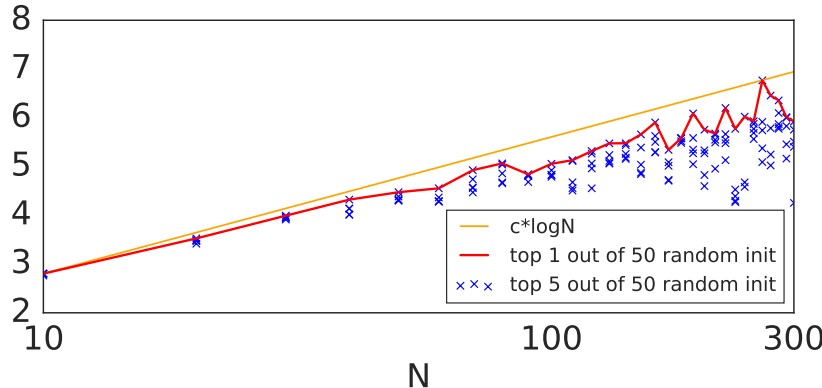

Figure 7: Lower bound on $\mathrm{Lip}_2(F)$ where $F$ is L2-MHA, with $D = 1$ and varying $N$, obtained by optimising $\|J_F(X)\|_2$ with respect to $X$, with 50 random initialisations of $X$ for each $N$.

In Figure 7, we show the lower bound on $\mathrm{Lip}_2(F)$ obtained by optimising $\|J_F(X)\|_2$ using the same optimisation procedure as for Figure 2 of Section 5.1. Here the optimisation is more difficult, evident in the variance of the top 5 values, and the trend is less clear, but it appears that $\mathrm{Lip}_2(f)$ grows at a rate of $O(\log N)$. The message is less clear here, and there are at least two possibilities:

(1) The optimisation is difficult even for small values of $N$, hence Figure 7 shows a loose lower bound.

(2) If the lower bound is tight, this suggests that the $O(\sqrt{N} \log N)$ bound in Theorem 3.2 is not asymptotically tight, and could be improved to $O(\log N)$ (or $O(\log N - \log \log N)$ as for $p = \infty$).

## L  OPTIMISING THE NORM OF THE JACOBIAN OF DP-MHA

In Figure 8, we show how the norm of the Jacobian $\|J_f(X)\|_\infty$ for `DP-MHA` $f$ keeps increasing when being optimised with respect to $X$. This is a useful sanity check validating our theoretical result of Theorem 3.1, that `DP-MHA` is *not* Lipshchitz. The oscillations are likely due to momentum term of Adam optimizer that was used to optimise the norm.

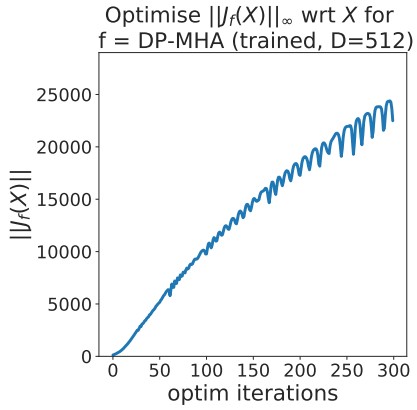

Figure 8: Optimise $\|J_f(X)\|_\infty$ w.r.t. $X$ for trained DP-MHA $f$.

# M  EXPERIMENT TYING KEYS AND QUERIES OF L2-MHA BUT PRESERVING PARAMETER COUNT

In Figure 4 of Section 5.3, we have shown that there is a clear reduction in performance when tying the keys and queries. To test whether this can be attributed to the reduction in parameter count, we tried doubling the number of columns of $W^Q$ when the keys and queries are shared (i.e. from $D/H$ to $2D/H$) so that the shared model has the same number of parameters as the unshared model. In Figure 9, the third column shows results for shared L2-MHA, but with the same number of parameters as the unshared L2-MHA i.e. without tying the keys and queries. The performance is similar to the second column (tying with a reduced number of parameters), suggesting that there is an inherent limitation in expressiveness to tying the keys and queries, and that the reduction in number of parameters is an insufficient explanation this phenomenon.

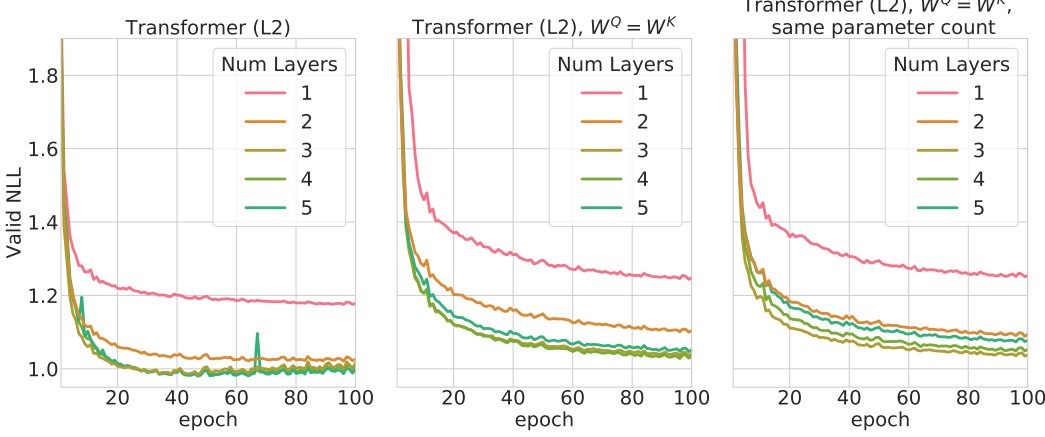

Figure 9: Experiment tying keys/queries but preserving parameter count.

# N  STABILITY EXPERIMENTS

In Figure 10 below, we compare the output variance of trained L2-MHA against trained DP-MHA, with weights from the one-layer Transformer (L2), $W^Q = W^K$ model and (DP) model used for Figure 4 respectively. We take the same distribution of inputs as used for the numerical invertibility experiment in Section 5.2, and show the histogram of inputs and outputs after flattening the input/output tensors. We see that the range of outputs remains similar to the range of inputs for Lipschitz L2-MHA, whereas for DP-MHA the outputs have a much wider range, because the Jacobian norm is large for DP-MHA at these inputs.

In practice, this leads to instabilities while training for DP-MHA, hence requiring careful tuning of the learning rate schedule for training deeper Transformer models: linear warmup and square root decay, as detailed in Appendix I. In Figure 11, we show how training instabilities arise for DP-MHA with deeper Transformer models if we use a **fixed learning rate**. We compare the training curves of DP-MHA, L2-MHA ($W^Q = W^K$) and Contractive-L2-MHA for a varying number of layers for the first 20 epochs of training with a fixed learning rate. We see that DP-MHA fails to train beyond 10 layers, whereas both L2-MHA ($W^Q = W^K$) (i.e. Lipschitz L2-MHA but not contractive) and Contractive-L2-MHA shows stable training for up to 18 layers. This was the deepest model we could fit on a single GPU, and we expect to be able to train even deeper models with L2-MHA ($W^Q = W^K$) and Contractive-L2-MHA. In Table 2 we show the best Test NLL across training of Transformer models with fixed learning rate. Note that for DP-MHA training becomes unstable beyond 10 layers, so we are only able to provide results up to 10 layers. The generalisation performance of the best model for each setting is similar.

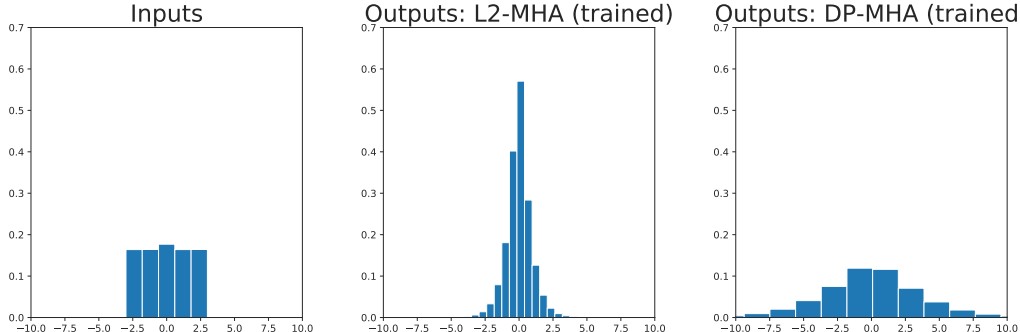

Figure 10: Histogram showing distribution of inputs/outputs of L2-MHA and DP-MHA

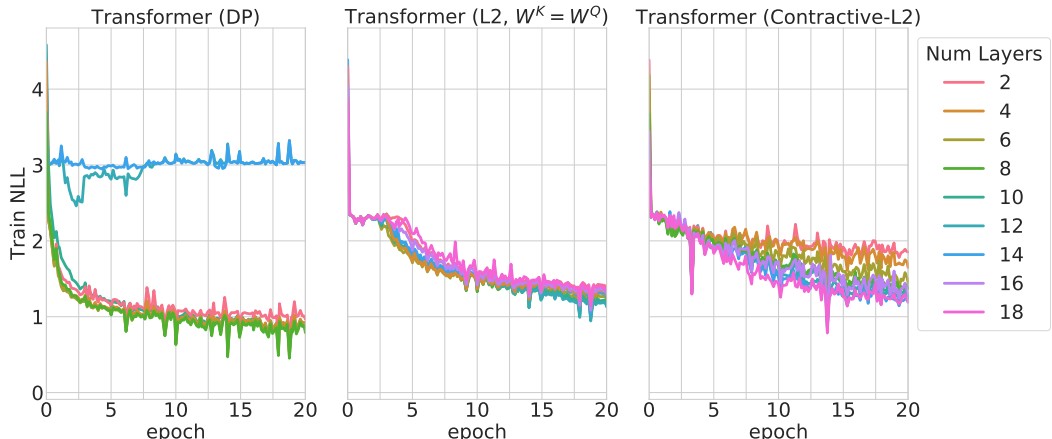

Figure 11: Train NLL for Transformer (DP), Transformer (L2) and Transformer (Contractive-L2)

| Number of Layers | 2 | 4 | 6 | 8 | 10 | 12 | 14 | 16 | 18 |
|---|---|---|---|---|---|---|---|---|---|
| **(DP)** | 1.061 | 1.032 | 1.021 | **1.017** | 1.025 | - | - | - | - |
| $W^Q = W^K$ **(L2)** | 1.168 | 1.040 | 1.023 | 1.024 | 1.019 | **1.008** | 1.018 | 1.027 | 1.034 |
| **(Contractive-L2)** | 1.246 | 1.135 | 1.103 | 1.079 | 1.072 | 1.060 | 1.039 | **1.029** | 1.031 |

Table 3: Test NLL for Transformer models trained with fixed learning rate on PTB character level language modelling

