# OpenReview forum: "The Lipschitz Constant of Self-Attention"
_ICLR.cc/2021/Conference — Reject_

### Official Review · AnonReviewer2 · 2020-10-20
**Official Blind Review #2**

**Rating:** 7
**Confidence:** 4

**Review:**

Summary:
This paper theoretically analyzes the Lipschitz constant of the self-attention module. In particular, the authors prove that the vanilla dot-product self-attention is not Lipschitz and propose a Lipschitz L2 self-attention whose Lipschitz constant is upper-bounded. The theoretical results and the asymptotic tightness of the derived bound are empirically verified by analytical experiments.

Reason for score:
I vote for an accept. I really like the theoretical analysis for the Lipschitz continuity of self-attention. And the proposed Lipschitz L2 self-attention has several potential practical applications. My major concern is whether the advantage of the Lipschitz L2 self-attention could outweigh the disadvantages in practice (see cons below). Overall, I think the theoretical contributions of this paper are sufficient for an accept.

Pros:
+ Studying the Lipschitz constant of self-attention is very interesting and motivated, since Lipschitz continuity is a desirable property for neural networks but most previous works only restrict to simple fully connected or convolution networks.
+ The proposed L2 self-attention is a simple plug-in variant of the standard dot-product self-attention and could potentially be applicable to several Lipschitz-required situations, e.g., the analysis of robustness and the design of invertible models.
+ Overall, the paper is clearly written and the derivations that I have checked are correct.

Cons:
- From the experiments, the expressivity of the proposed L2 self-attention is a big concern. It is not clear that the advantages of the Lipschitz L2 self-attention would outweigh its disadvantages. Therefore, it would be great to do more extensive experiments to indeed verify the superiority of Lipschitz continuity, e.g., the improved robustness and the training stability or the applicability to invertible models.
- In the experiments regarding the asymptotic tightness of the upper bound, only the setting where $H=D=1$ is shown. It would be more convincing to include experiments for larger dimensions as well.

Additional Comments/Questions:
1. Another potential application of the Lipschitz/Contractive L2 self-attention is DEQ[1] to ensure the uniqueness and the stability of fixed points. It would be great to include some discussions (if it is correct).
2. In Lemma E.1, is the equation between $\tilde{J}_{ij}$ and $\tilde{K}_{ij}$ correct?

[1] https://arxiv.org/abs/1909.01377

##########Post-Rebuttal Feedback########
I appreciate the author's thorough response and I think the additional experiments on robustness/stability make the paper stronger, so I decide to raise my score to 7. For the future version of the paper, it would be great to see more comprehensive experiment results that show the improved robustness/stability in the main text.

---

> ### Author Response · Authors · 2020-11-13
> **Author response**
>
> We would like to thank the reviewer for their time and effort in reading the paper to the extent of checking the derivations, and we are glad that you enjoyed the theoretical analysis. Below we address the concerns expressed in the review.
>
> *>It is not clear that the advantages of the Lipschitz L2 self-attention would outweigh its disadvantages. Therefore, it would be great to do more extensive experiments to indeed verify the superiority of Lipschitz continuity, e.g., the improved robustness and the training stability or the applicability to invertible models.*
>
> Indeed an aspect of Lipschitz L2 self-attention that addresses problems of the original DP self-attention is for robustness/training stability. We have revised the paper to include results that further investigate this. In Figure 10 of Appendix N, we investigate robustness by comparing the output variance of trained L2-MHA against trained DP-MHA. Using the same distribution of inputs used for the numerical invertibility experiment in Section 5.2, we show that the range of outputs remain similar to the range of inputs for Lipschitz L2-MHA, whereas for DP-MHA the outputs have a much wider range and higher variance, because the Jacobian norm is large for DP-MHA at these inputs.
> In practice, this can lead to training instabilities for DP-MHA, hence requiring careful tuning of Adam's learning rate schedule for training deeper Transformer models - linear warmup and square root decay, as detailed in Appendix I and used commonly in the literature. In Figure 11 of Appendix N, we show how training instabilities arise for DP-MHA with deeper Transformer models if we use a *fixed learning rate*  for Adam, whereas Contractive L2-MHA remains stable. We compare the training curves of Transformer(DP) and Transformer(Contractive-L2) for a varying number of layers. We see that Transformer(DP) fails to train beyond 10 layers, whereas Transformer(Contractive-L2) shows stable training up to 18 layers. This was the deepest model we could fit on a single GPU, and we expect to be able to train even deeper models with Contractive-L2-MHA.
>
> One notable example where it is absolutely necessary to have Lipschitz self-attention and non-Lipschitz self-attention fails is for residual flows [Berhmann et al '19, Chen et al '19]. If we would like to use self-attention/Transformers to form residual flows for density estimation, self-attention must be (globally) Lipschitz to guarantee invertibility. If we use the original dot-product MHA/Transformer that is not Lipschitz, the resulting model would not satisfy invertibility as shown in the experiments of Section 5.2, hence the density estimates that it gives would simply be incorrect. Given the theoretical focus of the paper, we believe an exploration of such challenging applications calls for a separate research project/paper that is more focused on the application. Hence we are currently working on a follow-up project that studies the application of L2 self-attention to residual flows for density estimation.
>
> *>In the experiments regarding the asymptotic tightness of the upper bound, only the setting where H=D=1 is shown. It would be more convincing to include experiments for larger dimensions as well.*
>
> We used this setting for computational constraints and ease of optimisation. Note that the input dimensionality is dim(X) = N*D, and we want to observe the behaviour of the lower bound with growing N (up to 1000). Hence for higher values of D, the optimisation problem max_X ||J_f(X)||_infty becomes difficult and computationally demanding (to use enough random initialisations for a thorough optimisation - as mentioned in Appendix I, we used 50 for each value of N to produce Figure 2). Note that for the lower bounds on the 2-norm Lipschitz constant obtained by max_X ||J_f(X)||_2 in Appendix K, there is evidence that the optimisation is difficult and leads to loose lower bounds, even for the D=H=1 setting.
>
> *>Another potential application of the Lipschitz/Contractive L2 self-attention is DEQ[1] to ensure the uniqueness and the stability of fixed points.*
>
> Thank you for this interesting reference - this is potentially related work we were not aware of. At a first glance, using 1-Lispchitz self-attention in the context of DEQ seems to imply the fixed-point iterations used for the forward pass will have a convergence guarantee. We will study it in more detail and include a discussion as necessary.
>
> *>In Lemma E.1, is the equation between $\tilde{J}_{ij}and\tilde{K}_{ij}$ correct?*
>
> Thank you for pointing this out. Indeed there is a typo in the first equation of the proof, where the right hand side should be right-multiplied by W^K^T. We have corrected this in the revised version of the paper. This does not affect the correctness of the proof as the rest of the proof remains valid.

---

> > ### Comment · AnonReviewer2 · 2020-11-23
> > **About the robustness/stability experiements**
> >
> > Thanks for providing more extensive experiment results regarding robustness/stability. I have some questions about these experiments:
> >
> > 1. It is not clear that the input/output distribution is directly related to the robustness of the model (though by intuition, it may be an indication of robustness). I would be very happy to see the evidence that shows the L2-attention indeed improves the robustness against adversarial attacks on NLP tasks.
> >
> > 2. In the experiment regarding training stability, it would be great to see the plots for L2-attention (but not contracting). Because in this setting, contractility may not be needed for training stability. And I am curious if L2-attention with Lipschitz constraint could match the performance of DP-attention while getting training stability.

---

> > > ### Author Response · Authors · 2020-11-23
> > > **Author response**
> > >
> > > Thank you for responding to our rebuttal and engaging in the discussion. We very much appreciate it. Here are our answers to your questions:
> > >
> > > 1. Regarding the robustness experiment, our choice of the input distribution can be seen as a form of adversarial attack, because these are precisely the inputs at which the Jacobian norm for the DP self-attention blows up (same as inputs used for the numerical invertibility experiment in Section 5.2). Of course this is a toy experiment that is only suitable for this short rebuttal period, and we believe that a more full-blown study of robustness against adversarial attacks for NLP models with Lipschitz constrained self-attention is a promising application for future research. Given the theoretical focus of our paper, we believe that such an application of Lipschitz constrained self-attention deserves a separate research project/paper that explores the application in depth.
> > >
> > > 2. Regarding the stability experiment, we have updated Figure 11 of Appendix N to include results for L2 self-attention ($W^Q=W^K$) that is Lipschitz but not contractive. Indeed we see that L2 self-attention also leads to stable training compared to DP self-attention, showing stable training up to 18 layers. As you have pointed out, the contractiveness doesn't seem necessary for stability in this particular problem. To compare generalisation performance of these models, we include Table 3, showing Test NLL for DP self-attention and L2 self-attention with fixed learning rate Adam optimisation. Note that for DP, training becomes unstable beyond 10 layers, hence we only show results up to 10 layers. We see that the best test-performances for the 3 choices of self-attention are very similar, showing that with fixed learning rates, (Contractive)-L2 can match the performance of DP due to more stable training with more layers.
> > >
> > > We thank you again for your comments and questions that have helped improve the paper, and look forward to your revised evaluation of the paper based on these discussions.

---

### Official Review · AnonReviewer3 · 2020-10-20
**A solid theoretical analysis of a state-of-the-art deep learning concept**

**Rating:** 7
**Confidence:** 2

**Review:**

The presented manuscript studies the Lipschitz constants of self-attention networks. It proofs that the widely used dot-product self-attention is not globally Lipschitz. The authors propose a so called L2 self-attention and derive its Lipschitz constant. An application in a transformer-based architecture shows only a minor performance reduction as price for the Lipschitz property.

The literature is properly reviewed and the contributions are clearly stated. The authors explain all concepts required to understand the theoretical derivation. However, I suggest that the difference between local and global Lipschitz continuity are introduced in detail as this becomes particularly important.
Furthermore, the proof of Theorem 3.1. was difficult to follow for me and I think there should be a simpler way to formulate it. A few suggestions: i)  first give a high level overview of the steps of the proof ii) use lemmas to show intermediate results iii) proof it for N=1, D=1 first, which will simplify the notation
One point, which is shortly discussed by the authors, is that the dot-product self-attention is local Lipschitz. For most applications, the inputs are naturally bounded e.g. images. i.e. also the DP-MHA is Lipschitz.  So it unclear for me whether the L2-MHA will be need in these cases and maybe an experimental comparison would be convincing here.
The experimental section evaluates the proposed approach from three quite distinct directions. While 5.1. relies quite heavily on empirical evidence to support a theoretical claim, the example 5.2. and 5.3. get a clear message across.
The supplementary material is excessive, but was not further considered for this review.

In general, I think the theoretical analysis of deep learning methods is important to advance the field and I am therefore in favor of this paper.

---

> ### Author Response · Authors · 2020-11-13
> **Author Response**
>
> We would like to thank the reviewer for their time and effort put into the review, and glad that you found the theoretical analysis in the paper important. Below we address the concerns expressed in the review.
>
> *>For most applications, the inputs are naturally bounded e.g. images. i.e. also the DP-MHA is Lipschitz. So it unclear for me whether the L2-MHA will be need in these cases and maybe an experimental comparison would be convincing here.*
>
> Indeed we discuss this point in Section 6, that can be summarised as: 1. Even if DP-MHA is Lipschitz on bounded input spaces, this fact is not helpful if we cannot give a (tight) bound on its Lipschitz constant - the true Lipschitz constant can be very large. Hence to make use of this fact we must bound the Lipschitz constant for DP-MHA, which would require non-trivial mathematical work. 2. Self-attention is typically applied at multiple layers within the Transformer, so the input to each self-attention layer will live in a different bounded set that depends on the parameters of previous layers, complicating the analysis.
>
> One aspect of Lipschitz L2-MHA that addresses problems of the original DP-MHA is for robustness/training stability. We have revised the paper to include results that further investigate this. In Figure 10 of Appendix N, we investigate robustness by comparing the output variance of trained L2-MHA against trained DP-MHA. Using the same distribution of inputs used for the numerical invertibility experiment in Section 5.2, we show that the range of outputs remain similar to the range of inputs for Lipschitz L2-MHA, whereas for DP-MHA the outputs have a much wider range and higher variance, because the Jacobian norm is large for DP-MHA at these inputs.
> In practice, this can lead to training instabilities for DP-MHA, hence requiring careful tuning of Adam's learning rate schedule for training deeper Transformer models - linear warmup and square root decay, as detailed in Appendix I and used commonly in the literature. In Figure 11 of Appendix N, we show how training instabilities arise for DP-MHA with deeper Transformer models if we use a *fixed learning rate*  for Adam, whereas L2-MHA (with $W^Q=W^K$, hence lipschitz but not contractive) & Contractive L2-MHA remains stable. We compare the training curves of Transformer DP, L2 ($W^Q=W^K$) and Contractive-L2 for a varying number of layers. We see that Transformer(DP) fails to train beyond 10 layers, whereas both Transformer (L2, $W^Q=W^K$) and Transformer(Contractive-L2) shows stable training up to 18 layers. This was the deepest model we could fit on a single GPU, and we expect to be able to train even deeper models with L2 and Contractive-L2-MHA.
>
> *>the proof of Theorem 3.1. was difficult to follow for me and I think there should be a simpler way to formulate it. A few suggestions: i) first give a high level overview of the steps of the proof ii) use lemmas to show intermediate results iii) proof it for N=1, D=1 first, which will simplify the notation*
>
> Thank you for the suggestion. As per your suggestion, we have reformatted the proof: we start off with a summary of the proof, and simplified the proof to the case D=1 and general N (since self-attention is not well-defined for N=1) and moved the general D,N case to the appendix. At the end of the proof, we have a paragraph giving high-level intuition of the result.

---

### Official Review · AnonReviewer4 · 2020-10-26
**Review Comments of Paper "The Lipschitz Constant of Self-Attention"**

**Rating:** 5
**Confidence:** 3

**Review:**

This paper shows that the widely used dot-product self-attention is not Lipschitz with respect to any p-norm with $p \ge 1$, and formulates L2 self-attention, which is Lipschitz.

The theorems and the proofs in the paper make sense to me, and the authors use experiments to show the tightness of Lipschitz constant derived from their theorem, the invertability using L2 self-attention, and the expressiveness of L2 self-attention on a NLP task compared with the transformer using the dot-product self-attention.

I think this paper is interesting, given that there is almost no theoretical paper (compared with fully connected NN and convolutional neural network) studying the attention mechanism of neural network.

However, my main concern is the “significance” of Lipschitz property. From a theoretical side, although there are many theoretical guarantees using the Lipschitz constant of the neural network or the general classification function, it may not lead to the best bounds. Actually, even if the original dot-product self-attention is not Lipschitz, I guess there is also some probability that most of the time (or along the trajectory) the self-attention is Lipschitz. This similar phenomenon happens in the Neural Tangent Kernel regime: although wide neural networks with ReLU have local minima and is not smooth everywhere, the function is easy to optimize along the trajectory.

From the experimental/application side, I am not so familiar with the invertible NN literature, and I am not so convinced that invertible NN can directly lead to the applications mentioned in the paper. I think the most straight-forward way (for me and general audience) to show the advantage L2 self-attention is to find some application, which may be easy and the experiments can be very preliminary, and show that L2 self-attention works better on these tasks in the preliminary experiments.

In the light of my previous concern, I would like to ask a few questions:
(1)	Do you find any cases through experiments that the dot-product self-attention does not perform well but L2 self-attention can perform better? E.g. with the L2 self-attention mechanism, the training of GAN is more stable than that using the original self-attention, or the adversarial training using L2 self-attention performs better than the original self-attention?

(2)	Is it very natural to extend the invertible NN to the tasks mentioned in the introduction, i.e. flow based generative models?

(3)	Are there any references claiming some lower bounds of the performance in terms of the Lipschitz constant? If yes, I suggest to add those references into a related work section, or the introduction, to make the contribution of this paper clear.

(4)	In the experiment, you mention that training the original transformer needs lots of tuning and the performance is not stable, while training transformer using L2 self-attention is much more stable. Could you please show more results on this phenomenon, because I think this phenomenon is interesting and can help to clarify question (1) I ask? E.g. the results of original transformer using different parameters with different initializations and the results of L2 transformer by choosing different parameters?

(5)	A suggestion: I think the proof of Theorem 3.1 can be put in the appendix and the conclusion can be more concise, and the main content can contain more experiment results and related works.


I will adjust my score after some of my questions are answered.

---

> ### Author Response · Authors · 2020-11-13
> **Author Response**
>
> We would like to thank the reviewer for their time and effort spent spent on this detailed review with constructive feedback. Below we address the concerns expressed in the review.
>
> *>(1) Do you find any cases through experiments that the dot-product self-attention does not perform well but L2 self-attention can perform better?(4) In the experiment, you mention that training the original transformer needs lots of tuning and the performance is not stable, while training transformer using L2 self-attention is much more stable. Could you please show more results on this phenomenon, because I think this phenomenon is interesting and can help to clarify question (1) I ask?*
>
> One aspect of Lipschitz L2 self-attention that addresses problems of the original DP self-attention is for robustness/training stability. We have revised the paper to include results that further investigate this. In Figure 10 of Appendix N, we investigate robustness by comparing the output variance of trained L2-MHA against trained DP-MHA. Using the same distribution of inputs used for the numerical invertibility experiment in Section 5.2, we show that the range of outputs remain similar to the range of inputs for Lipschitz L2-MHA, whereas for DP-MHA the outputs have a much wider range and higher variance, because the Jacobian norm is large for DP-MHA at these inputs.
> In practice, this can lead to training instabilities for DP-MHA, hence requiring careful tuning of Adam's learning rate schedule for training deeper Transformer models - linear warmup and square root decay, as detailed in Appendix I and used commonly in the literature. In Figure 11 of Appendix N, we show how training instabilities arise for DP-MHA with deeper Transformer models if we use a *fixed learning rate*  for Adam, whereas L2 & Contractive L2-MHA remains stable. We compare the training curves of Transformer DP, L2 ($W^Q=W^K$, hence Lipschitz but not necessarily contractive) and Contractive-L2 for a varying number of layers. We see that Transformer(DP) fails to train beyond 10 layers, whereas Transformer(L2, $W^Q=W^K$) and Transformer(Contractive-L2) show stable training up to 18 layers. This was the deepest model we could fit on a single GPU, and we expect to be able to train even deeper models with L2 and Contractive-L2-MHA.
>
> One notable example where it is absolutely necessary to have Lipschitz self-attention and non-Lipschitz self-attention fails is for residual flows [Berhmann et al '19, Chen et al '19]. This is a noteworthy setting for density estimation on sequence data given that Transformers have been dominant in tasks on sequence data across multiple data modalities, and residual flows are state-of-the-art on density estimation among normalizing flow models. If we would like to use self-attention/Transformers to form residual flows for density estimation, self-attention must be (globally) Lipschitz to guarantee invertibility. If we use the original dot-product MHA/Transformer that is not Lipschitz, the resulting model would not satisfy invertibility as shown in the experiments of Section 5.2, hence the density estimates that it gives would simply be incorrect. Given the theoretical focus of the paper, we believe an exploration of such challenging applications calls for a separate research project/paper that is more focused on the application. Hence we are currently working on a follow-up project that studies the application of L2 self-attention to residual flows for density estimation.
>
> *>(2) Is it very natural to extend the invertible NN to the tasks mentioned in the introduction, i.e. flow based generative models?*
>
> Yes, the use of invertible self-attention for residual flows [Berhmann et al '19, Chen et al '19] is very natural indeed, and the most natural application for us. Lipschitz self-attention can serve as a drop-in-replacement of the convolutions in the architectures proposed in these papers.
>
> *>(3) Are there any references claiming some lower bounds of the performance in terms of the Lipschitz constant?*
>
> In the introduction, we cite [Sokolic et al '17] as a reference that uses bounds on the Lipschitz constant (i.e. spectral norm of the Jacobian) to provide generalisation bounds of deep neural net classifiers. We will make this clearer in the paper.
>
> *>(5) A suggestion: I think the proof of Theorem 3.1 can be put in the appendix and the conclusion can be more concise, and the main content can contain more experiment results and related works.*
>
> Thank you for the suggestion. Given that the main contribution of the paper is theoretical, we thought it might be important to have some of the proofs of our theoretical results in the main paper. We have instead reformatted and simplified the proof of Thm 3.1 to cover the case of D=1, and moved the original proof covering general D to the appendix. We hope that this makes the proof more concise and easier to follow.

---

### Official Review · AnonReviewer5 · 2020-11-06
**Official Blind Review #1**

**Rating:** 5
**Confidence:** 4

**Review:**

This paper studies the Lipschitz continuity properties of self-attention. It is proved that the widely-used dot-product self-attention is not Lipschitz continuous. A novel L2 self-attention is proposed and proven to be Lipschitz continuous. Experiments show that the upper bound of Lipschitz constant for L2 self-attention is asymptotically tight. Invertibility of MHA residual map is investigated to prove the Lipschitz continuity of L2 self-attention. Finally, experiments on Transformers with L2 self-attention are studied.

The problem studied in this paper is a novel problem which is unexplored and interesting. The paper is equipped with solid mathematical analysis and experimental results. It is well written and easy to follow.

My main concern about this paper is --- why do we need Lipschitz continuity in self-attention. In the introduction, the authors mention some contexts where Lipschitz continuity properties of neural networks are needed. But for self-attention, what are the benefits of Lipschitz continuity, or are there any potential scenarios where non Lipschitz continuous self-attention models do not work but only Lipschitz continuous self-attention models do work. In the experiments of Transformer, the L2 self-attention model behaves just comparable (or sometimes worse) than DP self-attention, which is not a strong evidence of the necessity of Lipschitz continuity of self-attention models.

Minor questions:
1. The non Lipschitz continuity of DP attention models seem to come from the "quadratic" terms of X. L2 self-attention looks like Gaussian kernel in its form. I wonder is there any general assumptions (like using bounded kernel?) under which self-attention models are Lipschitz. This will probably generalize the mathematical content of this paper.

2. In Figure 4, could you provide the exact values of the valid NLL for each panel. It is difficult to distinguish the performance, especially the performance between panel 2 and panel 3, from the figures only.

---

> ### Author Response · Authors · 2020-11-13
> **Author Response**
>
> We would like to thank the reviewer for their time and effort spent on reviewing this paper. Below we address the concerns expressed in the review.
>
> *>why do we need Lipschitz continuity in self-attention...for self-attention, what are the benefits of Lipschitz continuity, or are there any potential scenarios where non Lipschitz continuous self-attention models do not work but only Lipschitz continuous self-attention models do work.*
>
> One notable example where it is absolutely necessary to have Lipschitz self-attention and non-Lipschitz self-attention fails is for residual flows [Berhmann et al '19, Chen et al '19]. This is a noteworthy setting for density estimation on sequence data given that Transformers have been dominant in tasks on sequence data across multiple data modalities, and residual flows are state-of-the-art on density estimation among normalizing flow models. If we would like to use self-attention/Transformers to form residual flows for density estimation, self-attention must be (globally) Lipschitz to guarantee invertibility. If we use the original dot-product MHA/Transformer that is not Lipschitz, the resulting model would not satisfy invertibility as shown in the experiments of Section 5.2, hence the density estimates that it gives would simply be incorrect. Given the theoretical focus of the paper, we believe an exploration of such challenging applications calls for a separate research project/paper that is more focused on the application. Hence we are currently working on a follow-up project that studies the application of L2 self-attention to residual flows for density estimation.
>
> Another aspect of Lipschitz L2 self-attention that addresses problems of the original DP self-attention is for robustness/training stability. We revised the paper to include results that further investigate this. In Figure 10 of Appendix N, we investigate robustness by comparing the output variance of trained L2-MHA against trained DP-MHA. Using the same distribution of inputs used for the numerical invertibility experiment in Section 5.2, we show that the range of outputs remain similar to the range of inputs for Lipschitz L2-MHA, whereas for DP-MHA the outputs have a much wider range and higher variance, because the Jacobian norm is large for DP-MHA at these inputs.
> In practice, this can lead to training instabilities for DP-MHA, requiring careful tuning of Adam's learning rate schedule for training deeper Transformer models - linear warmup and square root decay, as detailed in Appendix I and used commonly in the literature. In Figure 11 of Appendix N, we show how training instabilities arise for DP-MHA with deeper Transformer models if we use a *fixed learning rate*  for Adam, whereas L2 (with $W^Q=W^K$, hence Lipschitz) and Contractive L2-MHA remain stable. We compare the training curves of Transformer DP, L2 and Contractive-L2 for a varying number of layers. We see that Transformer(DP) fails to train beyond 10 layers, whereas Transformer(Contractive-L2) shows stable training up to 18 layers. This was the deepest model we could fit on a single GPU, and we expect to be able to train even deeper models with L2 and Contractive-L2-MHA.
>
> *>In the experiments of Transformer, the L2 self-attention model behaves just comparable (or sometimes worse) than DP self-attention, which is not a strong evidence of the necessity of Lipschitz continuity of self-attention models.*
>
> We would like to clarify that we are *not* claiming that L2 self-attention will be superior to DP self-attention in tasks where DP self-attention is already employed successfully. As mentioned at the beginning of Section 5.3, we expect that the Lipschitz constraint will limit expressiveness of self-attention, and the experiments in this section are for investigating by how much. Rather we are stating that dot-product self-attention is *not* suitable for applications that require Lipschitz constraints because it is *not* Lipschitz, and thus proposing a Lipschitz formulation of self-attention that *is* suitable.
>
> *>The non Lipschitz continuity of DP attention models seem to come from the "quadratic" terms of X ... I wonder is there any general assumptions (like using bounded kernel?) under which self-attention models are Lipschitz.*
>
> Thank you for this interesting suggestion. It would indeed be useful to generalise the L2-distance kernel to come up with a general assumption on the kernel that makes self-attention Lipschitz. However we think this is a difficult mathematical problem. We do not rely on boundedness when showing L2 self-attention is Lipschitz (in fact L2-distance (||(x_i - x_j)^T W^Q|| is unbounded in x_i,x_j), hence boundedness of the kernel is not a necessary condition for Lipschitzness, but it would be interesting to explore whether it is a sufficient condition.
>
> *>In Figure 4, could you provide the exact values of the valid NLL for each panel?*
>
> Yes. We have included Table 2 in Appendix I to give the exact NLL values in Figure 4.

---

### Decision · Program_Chairs · 2021-01-07
**Final Decision**

**Decision:**

Reject

**Comment:**

This paper shows that L2 self-attention is Lipschitz and presents a new method for computing the Lipschitz constant. All reviewers are positive about the technical part of the paper. However, the major concern comes from the significance of the computed Lipschitz constant. The paper only presents some numerical results using simple toy examples, which is insufficient to justify the  importance of the proposed method. The paper would be a much stronger paper if better numerical results could be presented.